# Type II innate lymphoid cell plasticity contributes to impaired reconstitution after allogeneic hematopoietic stem cell transplantation

Sonia J. Laurie [1,11], Joseph P. Foster II [1,2,11], Danny W. Bruce[1],
Hemamalini Bommiasamy[1], Oleg V. Kolupaev[1,10], Mostafa Yazdimamaghani[1],
Samantha G. Pattenden[3], Nelson J. Chao [4], Stefanie Sarantopoulos [4],
Joel S. Parker [1,5], Ian J. Davis[1,5,6] & Jonathan S. Serody [1,7,8,9] ✉

Type II innate lymphoid cells (ILC2s) maintain homeostasis and barrier integrity in mucosal tissues. In both mice and humans, ILC2s poorly reconstitute after allogeneic hematopoietic stem cell transplantation (allo-HSCT). Determining the mechanisms involved in their impaired reconstitution could improve transplant outcomes. By integrating single-cell chromatin and transcriptomic analyses of transplanted ILC2s, we identify a previously unreported population of converted ILC1-like cells in the mouse small intestine post-transplant. Exposure of ILC2s to proinflammatory cytokines resulted in a mixed ILC1-ILC2 phenotype but was able to convert only a small population of ILC2s to ILC1s, which were found post-transplant. Whereas ILC2s protected against acute graft-versus-host disease (aGVHD) mediated mortality, infusion of proinflammatory cytokine-exposed ILC2s accelerated aGvHD. Interestingly, murine ILC2 reconstitution post-HSCT is decreased in the presence of alloreactive T cells. Finally, peripheral blood cells from human patients with aGvHD have an altered ILC2-associated chromatin landscape compared to transplanted controls. These data demonstrate that following transplantation ILC2s convert to a pro-pathogenic population with an ILC1-like chromatin state and provide insights into the contribution of ILC plasticity to the impaired reconstitution of ILC2 cells, which is one of several potential mechanisms for the poor reconstitution of these important cells after allo-HSCT.

Innate lymphoid cells (ILCs) are haematopoietically derived cells in mucosal tissues poised to respond rapidly to inflammation and infection[1–3]. We demonstrated that conditioning therapy prior to murine allogeneic hematopoietic stem cell transplantation (allo-HSCT) leads to a marked decrease in ILC2s in the gastrointestinal (GI) tract and that the infusion of donor ILC2s improved the survival of recipient mice after transplant[4,5]. Importantly, ILC2s are also poorly reconstituted in patients following conditioning therapy and allo-HSCT[6,7]. Here, we evaluate the hypothesis that the absence of ILC2 cells in mice and humans after HSCT is mediated in part by the conversion of these cells to an alternate fate. Our data demonstrate that loss of these cells following transplantation is associated with conversion of those cells to an ILC1-like state and we identify regulators critical to this process.

## Results

To understand the fate of these cells after allo-HSCT, we first defined the chromatin state and transcriptome of a population of ILC2 cells generated ex vivo. Lineage-negative, ST2+ cells from the peritoneum and mesenteric lymph nodes of IL-25-treated mice were isolated, expanded in culture (Fig. 1a and Supplementary Fig. 1a), and compared to published ILC2s[4]. Histone 3 lysine 4 trimethylation (H3K4me3) signal at transcriptional start sites (TSSs), a chromatin mark of active transcription, strongly correlated with RNA abundance ($r = 0.81$)[4] (Fig. 1b). We identified ILC1 and ILC2-specific H3K4me3-marked TSSs from published data by differential analysis and hierarchical clustering (Supplementary Fig. 1b−e)[8]. H3K4me3 signal was significantly enriched at ILC2-associated compared to ILC1-associated TSSs (Fig. 1c). H3K4me3 was significantly enriched at the TSS of ILC2 lineage-defining transcription factors (TFs) and effector cytokines (Fig. 1d), whereas H3K4me3 was minimally enriched at the TSS of genes associated with ILC1 or ILC3 cells (Supplementary Fig. 1b)[9]. Assessment of TF expression and cytokine production in expanded ILC2s, revealed increased expression of GATA3 and IL-13, with no expression of Tbet or IFN-γ (Fig. 1e, f). Taken together, our IL-25 elicited, ex vivo expanded cells are epigenetically and phenotypically ILC2 cells.

To explore the hypothesis that the absence of ILC2 cells following allo-HSCT is a result of transdifferentiation mediated by cellular plasticity, we infused ex vivo expanded ILC2 cells from eGFP+ mice into lethally irradiated hosts reconstituted with donor BM cells and T cells at a ratio of ILC2 to T cells that significantly decreased acute graft-versus-host disease (aGVHD)[4,5] (Fig. 2a). After 14−21 days, eGFP+ cells could be recovered from recipient liver, lung, and small intestine (SI) (Supplementary Fig. 2a, b). We focused on donor ILC2 cells isolated

from the SI as the recovery of these cells is significantly impaired after allo-HSCT. We performed combined single nucleus (sn) RNA and ATAC (multiome) analysis on eGFP+ cells isolated from the mouse SI at day + 20, enabling us to evaluate the transcriptome and identify chromatin states of the same cell (Fig. 2a). We integrated pre- and post-transplant RNA abundance using canonical correlation analysis[10] and found that the preponderance of cells in the pre- and post-transplant populations segregated into two groups (Fig. 2b). Only 18.7% of post-transplant cells clustered with the pre-transplant ILC2 population. More impressively, very few pre-transplant cells clustered with the post-transplant population (0.3%), indicating that ILC2 cells isolated 20 days after transplantation were transcriptionally distinct from the infused cells.

One potential explanation for these findings is that the recovered post-transplant cells are not transdifferentiated from ILC2 cells but result from the expansion of a small population of contaminating ILC1-like cells in the pre-transplant population. As these cells represent less than 1% of the cells that were infused, they would have to markedly out proliferate the ILC2 cells to become the dominant population of cells in vivo. To evaluate this possibility, we marked ILC2 cells and ILC2 cells that had been treated with the proinflammatory cytokines IL-7, IL-12, IL-18, and IL-1β (pro-inflammatory cytokine conditioned ILC2s (pciLC2)) with distinct fluorescent markers⬚ If rapid expansion occurred, the pciLC2 cells would become the dominant population recovered in vivo even if only initially a small fraction of the input population. We transferred T cell depleted bone marrow along with one (1) tdTomato-expressing ILC1-like pciLC2 for every fifty (50) eGFP+ ILC2 together with an equal number of T cells (see Supplementary Fig. 3a). After 7−14 days, animals were

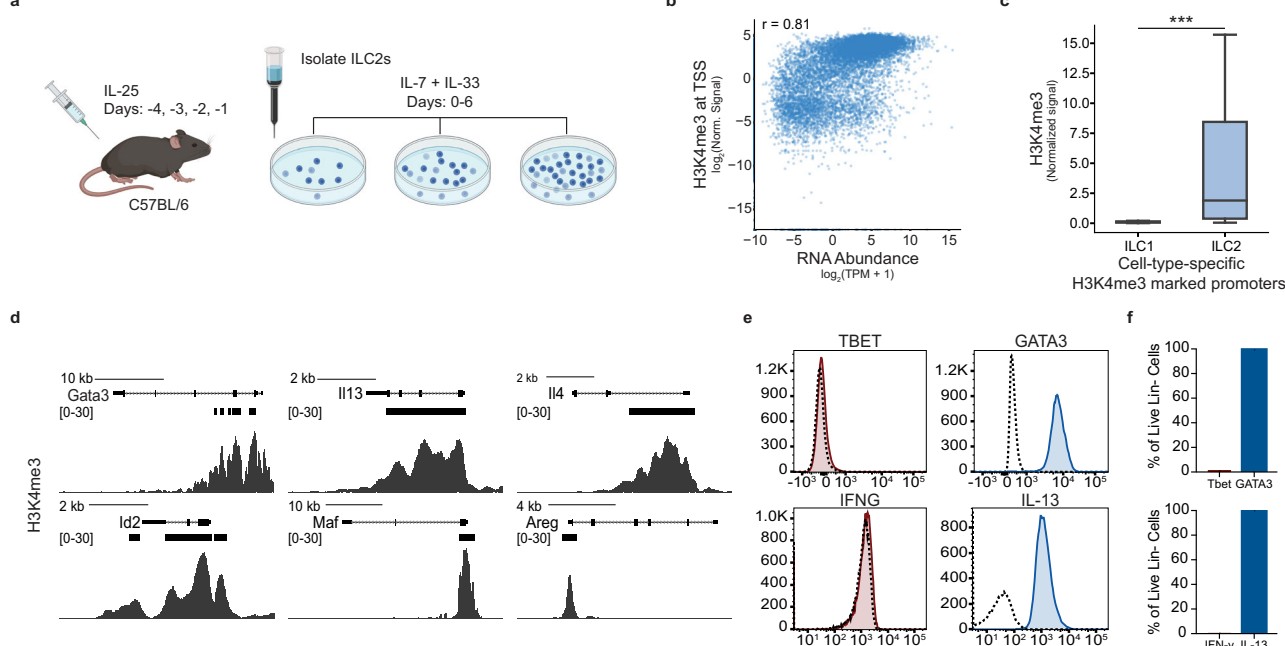

**Fig. 1 | Ex vivo expanded murine ILC2s exhibit the phenotypic and functional properties of ILC2s. a** Schematic of ex vivo ILC2 expansion. Mice received 4 days of IP IL-25 before the isolation of lineage-negative cells from the peritoneum and mesenteric lymph nodes. Cells were cultured for 6 days in complete media with IL-7 and IL-33. **b** Scatter plot of H3K4me3 signal from − 300 bp to +500 bp around the TSSs of all transcribed genes and RNA abundance of the corresponding gene (Spearman correlation coefficient, $r = 0.81$) TPM values represent the average signal of three replicates. H3K4me3 signal represents the average of two replicates. **c** Distributions of H3K4me3 signal in cells isolated from murine small intestine at TSSs that demonstrated differential H3K4me3 specific to ILC1 or ILC2 cells ($n = 2$, both conditions, one-sided Mann Whitney U Test, $p < = 5.37 \times 10^{-8}$). Box plot lines

represent min/max values. **d** Representative tracks of H3K4me3 signal at ILC2 lineage defining genes *Gata3*, *Il13*, *Il4*, *Id2*, *Maf*, and *Areg*. Black bars indicate regions of significant H3K4me3 enrichment (two-sided Wald Test (DESeq2), p adj < 0.05). **e, f** Following 6 days of expansion, ILC2s were stimulated and cells were characterized by multiparameter flow cytometry. Live, lineage-negative (CD3-, B220-, CD11b-, TER119-, Ly6G-) singlets were assessed for ST2, IFN-y, Tbet, GATA3, and IL-13 expression. **e, f** Show combined biological replicates where $n$ = minimum of three animals and represent two-three independent experiments. Dotted lines indicate unstimulated controls. Box plots in (**c**) are centered at the median. The bounds of each box represent the interquartile range IQR, and the whiskers represent the min and max values. Outliers are not shown in boxplots.

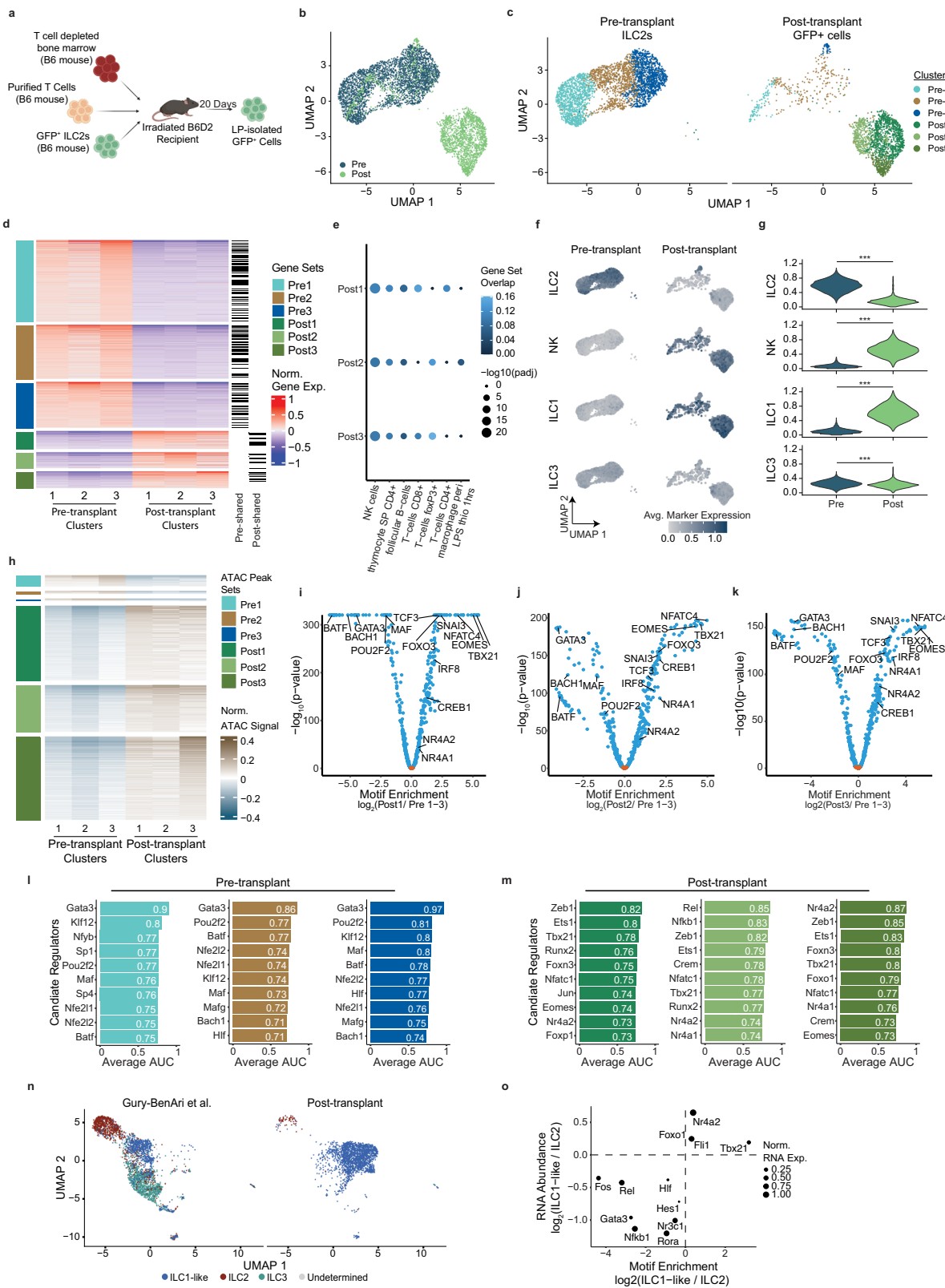

sacrificed and the accumulation of ILCs was quantified in GvHD target organs and the spleen (Supplementary Fig. 3b). We found approximately the same ratio of cells in each tissue as the initial input ratio demonstrating that ILC1-like cells do not outcompete ILC2s (Supplementary Fig. 3c–e). Next, the role of alloreactivity was evaluated by infusing marked donor ILC2 cells and TCD BM cells in haploidentical recipients in the absence of donor T cells or in

syngeneic recipients that receive T cell depleted BM cells, T cells and ILC2 cells from the recipient strain. Interestingly, there was a much greater expansion of donor ILC2 cells in the SI in syngeneic recipients compared with the very few ILC2 cells found in the SI in haploidentical recipients. These data suggest that the inflammatory environment post HSCT may limit ILC2 numbers in the SI and that donor T cells may be critical to allow donor ILC2 expansion.

**Fig. 2 | After allogeneic hematopoietic stem cell transplantation, a population of converted ILC1-like exILC2s emerges in the SI LP. a** Schematic of murine bone marrow transplantation model. Lethally irradiated B6D2 mice received allogeneic (C57BL/6) T cell depleted bone marrow plus total splenic T cells with activated GFP-expressing ILC2s. After 20 days, eGFP⁺ cells were isolated from the lamina propria (LP). **b** UMAP depicts embeddings of 5554 nuclei. Pre-transplant GFP⁺ ILC2s (blue) and post-transplant LP-isolated GFP⁺ cells (green) are shown. **c** Pre-transplant ILC2 and post-transplant cells are separately depicted. **d** Heatmap depicting differentially expressed genes (two-sided Wilcoxon rank sum test, p adj < 0.05, avglog2FC > 0) by comparing one pre-transplant cluster to the set of post-transplant clusters (Gene sets pre 1 – 3). Tick marks indicate genes shared across three pre-transplant gene sets (left) and post-transplant gene sets (right). **e** Gene ontology analysis was performed for each of the post-transplant associated gene sets (one-sided Fisher's Exact test). The color indicates the proportion of differential genes found in each ontology. **f** The average expression of marker genes associated with each type of ILC (ILC1, ILC2, ILC3, and NK)[11]. **g** Distribution of per cell average marker expression is shown. Pre signifies pre-transplant and post-indicates post-transplant (one-sided Mann Whitney U Test, ***p < 2.2 × 10⁻¹⁶).
**h** Regions of differential chromatin accessibility were identified between each pre- and post-transplant cluster from Fig. 2c. **i–k** Volcano plots showing the enrichment of DNA binding motifs at sites of increased chromatin accessibility in post-transplant clusters 1 (**i**), 2 (**j**), and 3 (**k**) relative to all the sites of open chromatin identified in the pre-transplant cell population. **l, m** Candidate regulators of pre- (**l**) and post- (**m**) transplant cells. **n** UMAP depicts a low-dimensional representation of the integrated RNA space, with each cell annotated with ILC cell type. **o** Scatter plot showing putative regulators of ILC1-like and ILC2s (Fig. 2n).

Having established that the post-transplant cells identified by 10x multiome sequencing are the result of transdifferentiation and not the outgrowth of a contaminating ILC1-like population, we next applied a shared nearest neighbor analysis to identify seven unique clusters of cells: three within the pre-transplant population and three within the post-transplant population (Fig. 2c). Differential expression analysis comparing each post-cluster to the entire pre-transplant population revealed similar genes across the three pre- and post-transplant clusters (Fig. 2d). To assess how the composition of these gene sets contributes to the gene expression profiles across the post-clusters, we calculated all possible intersections of the cluster associated gene sets. We found 276 genes shared across post-transplant Clusters 1–3 and 945 shared genes between the three pre-transplant clusters (Supplementary Fig. 2d). Genes induced in the three post-transplant sub-clusters were enriched for those associated with NK cells, an innate cell closely related to the ILC1 lineage (Fig. 2e). A seventh cluster of only two cells was not analyzed.

Next, we generated a transcriptional signature for each subset of ILC using published data and evaluated the expression of these genes[8]. Genes associated with ILC2 cells were most associated with the pre-transplant cells, although there was a small group of post-transplant cells with similar ILC2 gene expression. In contrast, NK- and ILC1-associated genes were most associated with the post-transplant eGFP⁺ cells (Fig. 2f, g). Interestingly, the post-transplant cells that clustered with the pre-transplant cells expressed the ILC1 signature genes, suggesting that following allo-HSCT, ILC2s can co-express genes of both the Type I and Type II transcriptional programs. ILC3-associated genes were minimally detected in both the pre- and post-transplant populations.

We next identified regions with differential chromatin accessibility (CA) between the pre- and post-transplant cells. We identified 24,990 sites of differential CA across the six clusters. CA was increased at 23,168 sites in post-transplant cell Clusters 1–3 compared with the pre-transplant populations (Fig. 2h). These regions were enriched for multiple TF motifs including ILC1-associated *Tbx21* and *Eomesodermin* (Fig. 2i–k). In contrast, sites with decreased CA were enriched for motifs of ILC2-defining TFs *Gata3, Maf, Batf*, and *Pouf2*[11,12].

To identify putative transcriptional regulators of cell states within each pre- and post-transplant cluster, we integrated gene expression with DNA motif enrichment, identifying 334 candidate regulators across six clusters. Pre-transplant clusters were associated with the ILC2 lineage-defining TFs *Gata3, Batf,* and *Pou2f2* (Fig. 2l), while post-transplant clusters were enriched for the ILC1 lineage defining gene *Tbx21* and the orphan nuclear receptor *Nr4a2* (Fig. 2m and Supplementary Fig. 2j, k). These findings were replicated using a second pair of samples (Supplementary Fig. 2g, h). To further assess ILC identity, we compared the transcriptomes of the post-transplant cells with CD127⁺ ILCs isolated from the small intestine of healthy mice[8]. Most post-transplant cells were annotated as ILC1-like, with a small number of cells maintaining an ILC2-like state (Fig. 2n and Supplementary Fig. 2k). Evaluation of differentially expressed transcription factors

whose binding motifs were enriched at sites of CA in the ILC1-like cell population revealed *Nr4a2*, *Foxo1*, and *Fli1* as potential candidate regulators, as well as the ILC1 master TF *Tbx21* (Fig. 2o and Supplementary Fig. 2l). Taken together, these data suggest that following transplantation, ILC2 cells exhibit plasticity, becoming ILC1-like with chromatin alterations at known (*Tbx21*) and potential regulators (*Nr4a2*, *Fli1*) at ILC1-specific genes.

Allo-HSCT conditioning regimens result in the release of pro-inflammatory cytokines including interleukin 6 (IL-6), IL-12, IL-23, and IL-1β[13–15]. Of these, IL-12 and IL-1β can regulate the plasticity of human ILC2s, leading us to hypothesize that the pro-inflammatory environment after allo-HSCT directs ILC2s towards an ILC1-like state[16–18]. We compared pcILC2s generated by culturing ILC2s with IL-7, IL-12, IL-18, and IL-1β, to ILC2s cultured in IL-7 and IL-33 alone (ILC2) (Fig. 3a). These conditions generated ILC2s and pcILC2s that remained lineage negative, with decreased expression of ST2 in pcILC2s (Supplementary Fig. 4a, b). In the pcILC2s, expression of Tbet and IFN-γ was detectible in a small fraction of cells where Tbet-positive cells retained Gata3 expression, while IFN-γ secretion was associated with decreased IL-13 (Fig. 3b, c). Targeted mRNA analysis of bulk pcILC2s revealed an increase in expression of *Tbx21*, *Ifng*, *Fasl*, and *Stat1*, and decreased *Il13, Il5*, and *Tnfsfr8* transcripts (Fig. 3d). Because changes in chromatin state may precede transcriptional and protein differences, we assessed CA and observed alterations at 19,201 regions after exposure to pro-inflammatory cytokines (Fig. 3e). Increased CA was found at ILC1 and ILC2 lineage defining genes (*p* < 0.05), with decreases at the *Il13* locus and increase near the *Ifng* TSS (Fig. 3f and Supplementary Fig. 4c).

Because exposure of in vitro expanded ILC2s to cytokine alone was not sufficient to transdifferentiate the majority of ILC2 cells, we evaluated other inflammatory mediators. Culture of ILC2 cells with resiquimod, a TLR7/8 agonist, or HMGB-1, a damage-associated molecular pattern motif-containing molecule released by dying cells, did not affect the survival of treated and untreated cells or their lineage negativity (Supplementary Fig. 5d). The addition of these transplant-relevant stimuli also did not alter the expression of Tbet or secretion of IFN-g, (Supplementary Fig. 4e, f).

Next, we performed combined snRNA and snATAC analysis of our in vitro cytokine-conditioned cells. We observed that ILC2 and pcILC2 segregated into two groups, which was consistent across replicates (Fig. 3g and Supplementary Fig. 4g). We calculated the aggregate CA at published genes associated with ILC1 (n = 347) and ILC2s (n = 422)[8]. There was a decrease in CA at ILC2-associated genes (Fig. 3h) but not an increase at ILC1 genes (Supplementary Fig. 4h), suggesting an intermediate chromatin state. We then asked if cytokine exposure produced changes in gene expression suggestive of an ILC1-ILC2 intermediate cell state by evaluating the expression of genes previously associated with both ILC1 and ILC2 cells. We found cytokine exposure significantly increased the expression of these genes compared to ILC2s with decreased expression of ILC2-associated genes (Fig. 3i). We also observed a decrease in gene expression at ILC1 genes and genes shared between ILC1/3 s, suggesting cytokine-mediated

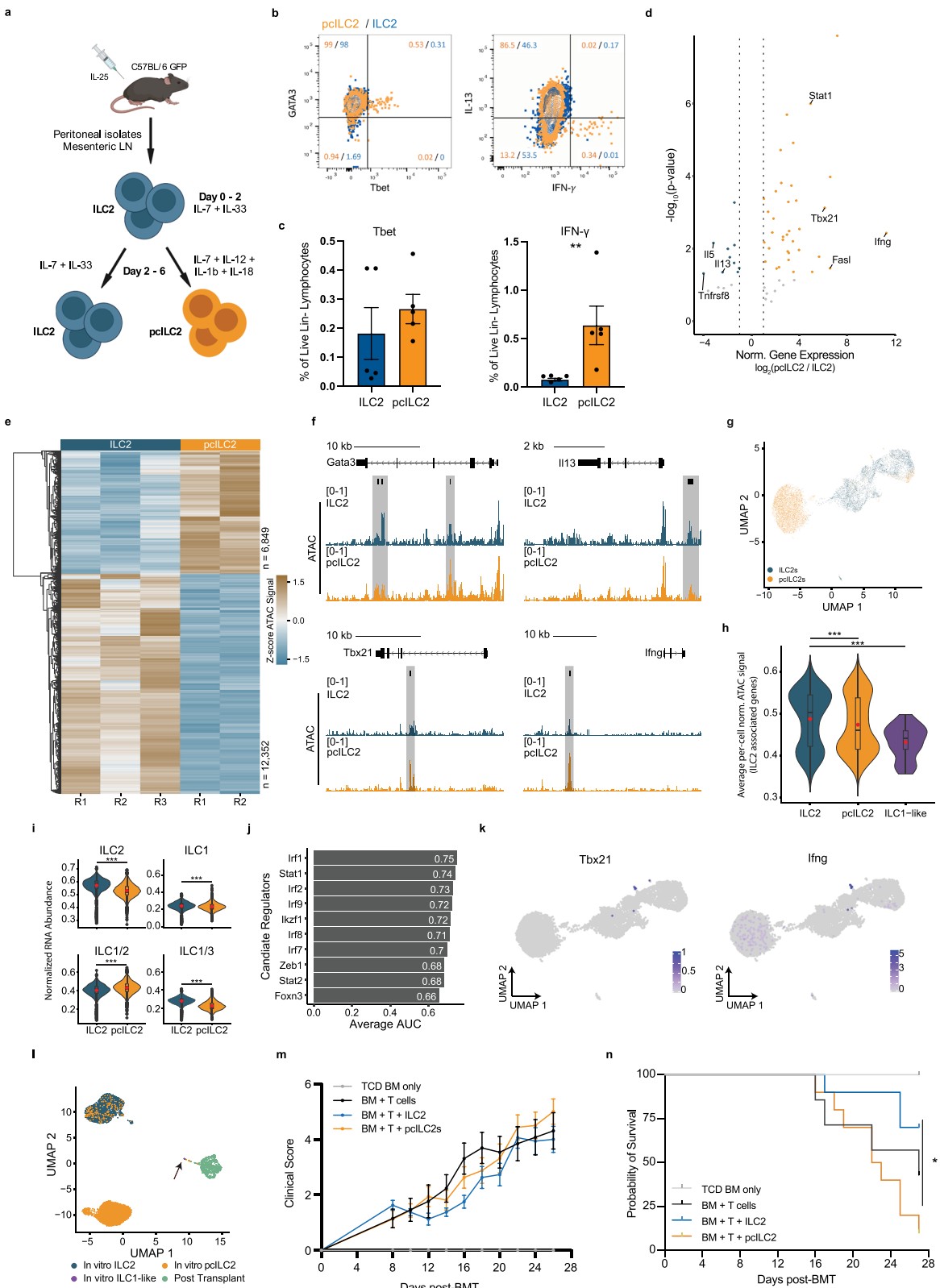

skewing of ILC2s induces an intermediate ILC1/2 cell state. Cytokine exposure was associated with differential expression of 11,389 genes, including decreased ILC2-associated *Il13* and *Il15* and increased *Bcl2*, *Stat1* and *Inpp4b*, genes associated with an ILC1-like transcriptional program[19–21] (Supplementary Fig. 4i). DNA motifs enriched at sites of differential CA following cytokine treatment included ILC1-associated *Irf2* and ILC2-associated *Irf4* and *Irf7*, as well as *Irf8*, which is known to

be downregulated as ILC precursors undergo lineage commitment (Supplementary Fig. 4j)[12,22,23]. To determine transcriptional regulators that may mediate this intermediate state, we merged TF gene expression with DNA binding motifs enriched at sites of differential CA and identified *Stat1* and several members of the *Irf* gene family (Fig. 3j). Unsupervised clustering revealed a small population of *Tbx21* and *Ifng* expressing cells, consistent with flow cytometric analyses that also

**Fig. 3 | In vitro skewing with IL-12 mediates ILC2 conversion resulting in the generation of a population of ILC1-like ILC2s. a** Schematic depicting ILC2s elicitation and isolation. **b, c** Representative flow cytometry plots of intracellular cytokine (**b**) and transcription factor expression (**c**) from cells expanded in vitro as described in 3 A. Error bars in (**c**) represent SEM. **d** ILC gene expression profiles based on mRNA abundance. **e** Differential sites of CA (two-sided Wald Test DESeq2), p adj < 0.05) between ILC2 and pcILC2s. The heatmap displays a z-score normalized ATAC signal. R1 and R2 indicate replicate numbers. **f** Representative tracks of normalized ATAC signal. Tick marks indicate differential sites of CA (two-sided Wald Test (DESeq2), p adj < 0.05). kb = kilobase. **g** UMAP of integrated RNA abundance signal from 7918 nuclei. ILC2s cultured with IL-7 and IL-33 (blue), ILC2s grown with proinflammatory cytokines, pcILC2 (orange). **h** Average chromatin accessibility at murine ILC2 marker genes[8].The Red dot shows mean value, ILC2-pcILC2: $p < 2.2 \times 10^{-16}$, ILC2-ILC1-like: $p = 7.46 \times 10^{-5}$. **i** Average per cell expression of

genes associated with ILC1, ILC2, shared between ILC1/2, and ILC1/3[8]. The analysis includes 4,758 ILC2 and 5641 pcILC2 nuclei. ***$p < 2.2 \times 10^{-16}$. **j** Candidate regulators of pcILC2 cells. **k** UMAP of integrated ILC2 and pcILC2s annotated with normalized gene expression. **l** UMAP of weighted nearest neighbor integration of single nucleus RNA and ATAC from in vitro ILC2s, pcILC2s, and post-transplant cells. The black arrow highlights ILC1-like cells. **m, n** Lethally irradiated B6D2 mice received T cell depleted bone marrow (BM, BM only), BM plus total splenic T cells (BM + T cells), BM plus T cells with activated ILC2s (BM, T cells + ILC2) or pcILC2s (BM, T cells + pcILC2). Mean ± SEM, $n = 7$–16. **n** Kaplan-Meier survival curve after allo-HSCT. Representative of 2 independent experiments with $n = 6$–10 mice. Log-rank (Mantel-Cox) test, ***$P < 0.01$. Boxplots (**h, i**), are centered at median. Box bounds represent the interquartile range (IQR), whiskers represent the min/max values, and outliers are not shown.

identified a small number of Tbet[+] IFN-y-secreting cells (Fig. 3k and Supplementary Fig. 4k). This population was also characterized by *Gzmb* expression and enrichment of Tbx21 motif and decreased and increased aggregate CA at ILC2 and ILC1 genes, respectively (Supplementary Fig. 4l, m). The weighted nearest neighbor analysis demonstrated that this population segregated from most pcILC2s but clustered near the eGFP[+] ILC1-like cells isolated following allo-HSCT (Fig. 3l). Taken together, these data demonstrate that proinflammatory cytokines induced a mixed ILC1/ILC2 gene expression pattern, with a small population that had CA and transcriptional expression similar to the ILC1-like cells that were isolated following transplantation. This limited conversion may reflect the brief cytokine exposure in the in vitro system or the absence of factors specific to the in vivo tissue microenvironment of the GI tract in GvHD. To determine the effect of pcILC2s in an aGvHD model, we infused lethally irradiated B6D2 mice with bone marrow alone or bone marrow and T cells together with either ILC2 or pcILC2s. Strikingly, pcILC2s failed to mitigate aGVHD and instead led to an increase in disease mortality (Fig. 3m, n).

We then evaluated whether chromatin state alterations identified in our murine model could also be detected in patients undergoing allo-HSCT. In patients, ILC2 conversion could occur in circulating cells in the bloodstream, in tissues, or at both sites. We evaluated the peripheral blood compartment for CA signal at regions associated with human ILC2 (hILC2) in patients following allo-HSCT. To identify CA alterations associated with cytokine treatment of human ILC2 cells, we performed ATAC-seq from nuclei of ILC2s enriched via RosetteSep separation from healthy donor peripheral blood mononuclear cells (PBMCs) and grown in culture with IL-2, IL-7, IL-33, IL-25, and IL-4 (hILC2) or with proinflammatory cytokines IL-12, IL-1β and IL-18 added to IL-2 and IL-7 (pc-hILC2) (Supplementary Fig. 5a, b and Fig. 4a). We identified 7456 sites of CA specific to the hILC2s, and 5051 sites specific to the pc-hILC2s (Fig. 4b). Among the TF motifs that were enriched in these sites, we identified BATF in the hILC2, and RORA and TBX21 in the pc-hILC2 (Fig. 4c). hILC2s demonstrated a significantly greater ATAC signal at GATA3 sites, whereas pc-hILC2s exhibited a greater signal at TBX21 sites (Fig. 4d). These changes suggest a shift in the chromatin state from ILC2- to ILC1-like in human cells following proinflammatory cytokine conditioning.

We then performed snATAC-seq on peripheral blood cells from six adult allo-HSCT recipients, half of whom were diagnosed with aGVHD in the first 100 days post-transplant and half of whom did not develop aGVHD during this time period (see Supplementary Data 1). Host-derived PBMCs were collected prior to transplantation and donor-derived PBMCs were collected post-transplantation at the time of aGVHD diagnosis (~ 4–6 weeks), or three months post-transplant in stable recipients. PBMCs from two healthy donors and one sample of in vitro expanded hILC2s were also analyzed (Fig. 4e). We identified genes with aggregate CA that differed between pre- and post-allo-HSCT. ATAC signal was significantly increased at 643 genes in the pre-

transplant samples, and 356 genes following transplant (Fig. 4f). Genes with higher signal in pre-transplant samples were associated with T cell activation and hematopoiesis, whereas genes with higher signal in post-transplant samples were associated with metabolic processes and positive regulation of cell motility and migration. (Fig. 4g). We then calculated the per cell average ATAC signal for each sample at sites that had been specifically associated with the expanded hILC2s in our bulk ATAC-seq data (Fig. 4b). As expected, we found that the highest average hILC2 signal was detected across the population of expanded hILC2s. Strikingly, we observed a significant decrease in aggregate ILC2-associated signal in the cells collected from post-transplant patients with aGvHD compared to those without aGvHD ($p$-value < 2.2e-16, Fig. 4h and Supplementary Fig. 5c). We repeated this analysis using the pc-hILC2 associated regions (see Fig. 4b), noting that the average signal across the cell population was increased in all post-transplant patients ($p$-value < 2.2e-16), with the greatest signal in individuals without aGvHD (Fig. 4i). Next, we evaluated the single published single-cell transcriptomic dataset post HSCT[24]. ILC2 cells were found in a subset of colon biopsies from a healthy patient but were not present in those with acute GvHD; however, the limited number of samples available greatly limited conclusions from these data. Taken together, CA data suggest that the state of peripheral blood cells is altered following allo-HSCT and that cells persisting after transplant in patients with aGVHD have a reduction in CA at ILC2-associated sites.

## Discussion

Following allo-HSCT human ILC2 cells do not recover to pre-transplant levels, despite the presence of ILC2 precursors in the blood, and the impaired reconstitution of ILC2 cells is associated with poor outcomes following transplantation[6]. Here, we show that ILC2 conversion occurs in mice after allo-HSCT with over 80% of infused ILC2 cells acquiring an ILC1-associated transcriptome and chromatin landscape after 20 days (Fig. 2). In vitro culture of ILC2s with transplant-associated proinflammatory cytokines, resulted in transcriptome and chromatin state changes associated with STAT1 and IFN-γ signaling, and infusion of these pcILC2 cells exacerbated acute GVHD in a murine HSCT model (Fig. 3). Finally, we demonstrate that following allo-HSCT, blood cells from patients with aGvHD have decreased CA signal associated with ILC2 cells, and an increase in CA signal for pILC2 cells that was independent of a diagnosis of aGvHD, (Fig. 4).

Based on our analysis of transitional and fully converted ILC2s, *Nr4a2* and *Fli1* may be potential regulators of ILC2 plasticity governing the emergence of this ILC1-like population, as both DNA binding motifs associated with these transcription factors were highly enriched in vivo converted ILC1-like cells, with NR4A2 expression also detected in our ILC1-like pcILC2 population after in vitro expansion. NR4A2 is an orphan nuclear receptor that negatively regulates T cell responses and is expressed by NK cells and ILC1s in the small intestine[11,25]. Although expression of FLI1, a member of the ETS family of winged helix-turn-helix transcription factors, has also been described in ILC1s[12], a role for

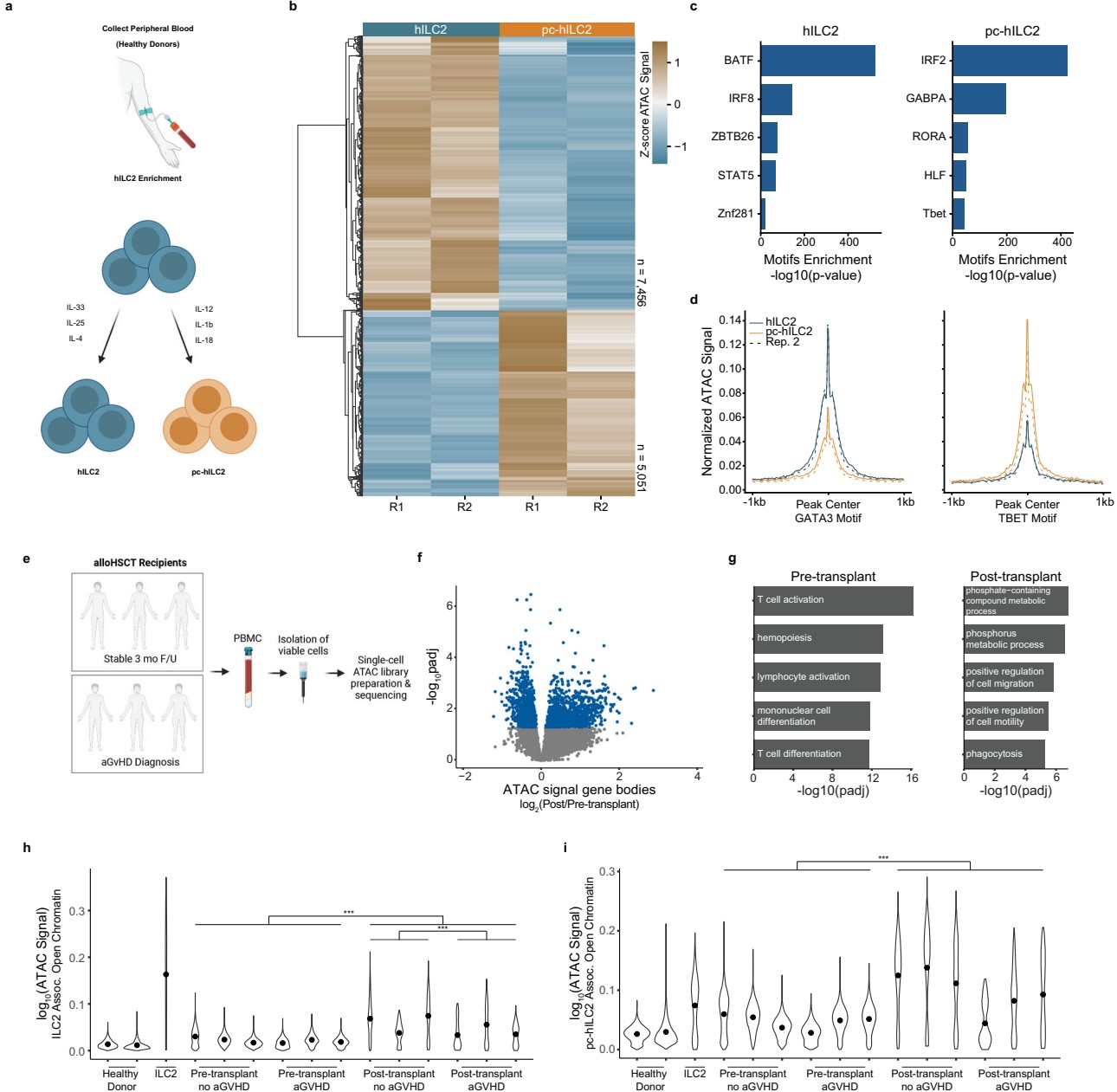

**Fig. 4 | Following allo-HSCT, blood cells from patients have decreased CA at sites of accessibility associated with ILC2 cells at the time of the diagnosis of aGVHD. a** Isolation of human ILC2s (hILC2s) from peripheral blood of healthy donors. Enrichment of ILC2s is followed by expansion of ILC2s and transdifferentiation for 14–21 days with IL-7 and IL-2 and IL-4, IL-33, IL-25 for hILC2s or IL-12, IL-18, IL-1β for proinflammatory cytokine human ILC2s (pc-hILC2s). **b** ATAC heatmap of z-score normalized ATAC signal of differential regions (two-sided Wald Test DESeq2), p adj < 0.05) of chromatin accessibility. R1 and R2 indicate replicate numbers. **c** Enriched motifs identified at differential sites of CA in hILC2 and pc-hILC2s (HOMER). **d** Average normalized ATAC signal was calculated at differential regions of chromatin accessibility that contain the GATA3 (left) or Tbet (right) binding motif. Dashed lines represent a second replicate. Rep 2 indicates replicate 2 (**e**) Blood was collected from patients 2–28 days prior to allogeneic HSCT. Post-transplant samples were collected at the time of aGVHD diagnosis (disease) or during a scheduled three-month follow-up (stable). Frozen buffy coats were thawed, and live cells were isolated by magnetic separation prior to snATAC-seq. **f** Differential analysis of aggregate ATAC signal over gene bodies +/− 2 kb for all genes. Volcano plot depicting log2FoldChange between pre- and post- samples for each gene score and − log10(p adj). Blue dots indicate p adj < 0.05 (two-sided Wald Test (DESeq2), $p < 0.05$). **g** Gene ontology (GO) analysis was performed for genes with differential ATAC signals for both the pre- and post-transplant cells. Bar plots indicate enrichment (− log10(p adj)) for the top 5 terms (One-sided Fisher's Test). **h** Average normalized ATAC signal at hILC2-associated sites of CAs (Fig. 4b) was calculated per nucleus for each patient sample. Violin plots show the top 90% of the signal. Black dots indicate the mean of each distribution. **i** Average normalized ATAC signal at pc-hILC2-associated sites of CAs (Fig. 4b) per nucleus for each patient sample. Violin plots show the top 90% of the signal. Black dots indicate the mean of each distribution (***$p < 0.001$, two-sided $t$ test).

NR4A2 or FLI1 in regulating ILC plasticity has not previously been demonstrated. Supporting our hypothesis that the plasticity of ILC2s is one of several potential mechanisms for the poor reconstitution of these important cells after allo-HSCT, we found a significant reduction in ILC cells in patients post-transplant that limited our ability to perform multiomic evaluation on individual samples. Thus, for human samples, the ATAC evaluation was performed on pooled PBMCs isolated after transplant. These data complement previous work that showed impaired reconstitution of NCR+ ILC3 cells after clinical allo-HSCT, with murine studies demonstrating that ILC3 cells in the GI tract

can mitigate acute GvHD of the GI tract via the expression of IL-22[25]. In summary, we demonstrate that loss of ILC2 cells after allogeneic stem cell transplant is associated with the conversion of those cells to an ILC1-like phenotype and have identified potential regulators that may be critical to this process.

## Methods

### Study approval
All experiments were performed in accordance with protocols approved by the University of North Carolina Institutional Animal Care and Use Committee (application number 14-001). Prior to sample collection, all human patients signed informed consent under Duke University IRB study protocol Pro00110250. Healthy donor control cells were purchased from Memorial Blood Center (St. Paul, MN) where all commercially available products are collected with IRB approval or exemption.

### Mice
C57BL/6 (strain 0000664) and C57BL/6 J × DBA/2 F1 (B6D2, strain 100006) mice were purchased from The Jackson Laboratory. The generation of enhanced GFP-expressing C57BL/6 mice has been described previously[26]. Donor and recipient mice were age-matched males between 8 and 16 weeks. Animals were housed under specific pathogen-free (SPF) conditions on a 12-h dark/light cycle at 21–22 °C and 30–70% humidity. Where applicable in all mouse studies, animals were euthanized via CO2-compressed carbon dioxide gas in cylinders followed by physical cervical dislocation to ensure death.

### Isolation of murine ILC2s
Eight- to 16-week-old B6 mice were given 0.4 µg recombinant mouse IL-17E/IL-25 (R&D Systems) by i.p. injection for 4 days. On day 5, cells were isolated from the mesenteric lymph nodes and peritoneum by peritoneal lavage using RPMI-1640 supplemented with 10% FBS, 2 mM L-glutamine, 12 mM HEPES, 0.1 mM non-essential amino acids, 1 mM sodium pyruvate, 1% Pen/Strep, and 50 µM 2-mercaptoethanol (complete media). ILC2s were isolated by negative selection with a MACS column using the following antibodies (anti-CD8α [clone 53-6.7], anti-CD4 [RM 4.4], anti-CD3ε [clone 145-2C11], anti-γδTCR [UC7-13DS], anti-TER119 [TER-119], anti-B220 [RA3-6B2], anti-CD11b [M1/70], anti-NK1.1 [PK136], eBioscience; anti-CD11c [N418], anti-CD19 [MB19-1], anti-Ly6G [1A8], and anti-CD49b [DX5], BioLegend) and Streptavidin Microbeads (Miltenyi 130-048-101). Cells were expanded in culture at 2.25 x 10^5 cells/mL in 24 well flat bottom TC-treated plates or flasks (Corning) as described below.

### Ex vivo expansion of murine ILC2s in culture
ILC2s were cultured at 2.25 x 10^5 cells/mL for 6 days in complete media (RPMI-1640 supplemented with 10% FBS, 2 mM L-glutamine, 12 mM HEPES, 0.1 mM non-essential amino acids, 1 mM sodium pyruvate, 1% Pen/Strep, and 50 µM 2-mercaptoethanol) and supplemented with 10 ng/ml rIL-7 and rIL-33 (PeproTech), with the media changed every 2 days. ILC2 activation was evaluated using flow cytometry on day 6 by surface and intracellular cytokine staining with antibodies against Lineage (eBioscience) and ST2 (Thermo Scientific). For experiments in which cells were generated via cytokine-mediated skewing (pcILC2s), cells were cultured at 2.25 x 10^5 cells/mL for 48 h in complete media supplemented with 10 ng/ml rIL-7 and rIL-33 (PeproTech). On Days 2 and 4, the media was replaced with complete R10 containing 10 ng/ml rIL-7, 10 ng/ml rIL-12, 10 ng/ml rIL-1b, 10 ng/ml rIL-15, 10 ng/ml rIL-2, and 5 ng/ml rIL-18.

### Isolation of human ILC2s from peripheral blood
Non-mobilized healthy donor peripheral blood (PB) leukapheresis products were purchased from Memorial Blood Center (St. Paul, MN), and human ILC2 cells were captured via the RosetteSep human ILC2 enrichment kit (STEMCELL Technologies), per the manufacturer's instructions.

### Ex vivo expansion of human ILC2s in culture
ILC2s were cultured at 2.25 x 10^5 cells/mL for 6 days in complete media (RPMI-1640 supplemented with 10% FBS, 2 mM L-glutamine, 12 mM HEPES, 0.1 mM non-essential amino acids, 1 mM sodium pyruvate, 1% Pen/Strep, and 50 µM 2-mercaptoethanol) and supplemented with 10 ng/ml rIL-7 and rIL-33 (PeproTech), with the media changed every 2 days. ILC2 activation was evaluated using flow cytometry on day 6 by surface and intracellular cytokine staining with antibodies against Lineage (eBioscience) and ST2 (Thermo Scientific). For experiments in which cells were generated via cytokine-mediated skewing (pcILC2s), cells were cultured at 2.25 x 10^5 cells/mL for 48 h in complete media supplemented with 10 ng/ml rIL-7 and rIL-33 (PeproTech). On Days 2 and 4, the media was replaced with complete R10 containing 10 ng/ml rIL-7, 10 ng/ml rIL-12, 10 ng/ml rIL-1b, 10 ng/ml rIL-15, 10 ng/ml rIL-2, and 5 ng/ml rIL-18.

### Flow cytometric & FACS analysis
The phenotype and function of murine ILC2s were evaluated by flow cytometry with antibodies as listed in Supplementary Data 2. Prior to transplantation T cells were evaluated by surface staining of CD4 (GK1.5) and CD8 (clone 53-6.7). Sample acquisition was performed using a BD LSRII or BD LSRFortessa (BD Bioscience) or a MACS Quant (Miltenyi Biotec), and data were analyzed by FlowJo v9/10 (TreeStar, BD) and Prism v10 (GraphPad).

### Transplantation models
Total T cells were isolated using a Cedarlane T cell recovery column kit (Cedarlane Laboratories), followed by antibody depletion using PE-conjugated anti-mouse B220 (RA3-B62) and anti-mouse CD25 (3C7) antibodies (eBioscience) and magnetic bead selection using anti-PE beads (130-0480801, Miltenyi Biotec). TCD bone marrow was prepared as described previously[27]. The day prior to transplantation, recipient mice received 950 cGy of total body irradiation. Recipients were intravenously injected with 4 × 10^6 T cells and 3 x 10^6 TCD BM cells. For ILC2 treatment groups, B6D2 recipients also received 3 – 4 × 10^6 ILC2s, respectively. Recipients were monitored three times a week and scored for clinical GVHD symptoms (designated "clinical score") using a semiquantitative scoring system as previously described[28,29]; animals were coded for these evaluations. The sample size was chosen for the effect size needed based on our previous experience with sample sizes needed to demonstrate a significant difference in GVHD scoring between control and treated groups. For the scoring evaluation experiments, the inclusion of 9–12 recipients provided a power of 90% to detect a difference of 14 days in the median GVHD score of ≥ 5 with an α error of < 0.05 between control and treated groups. For all experiments, a control group received TCD bone marrow alone without additional T cells, which controlled for the presence of T cells in the marrow inoculum and potential infectious complications during aplasia.

### Cell isolation from GVHD target and control organs
Animals were euthanized with CO$_2$ followed by cervical dislocation, and spleen, liver, lungs, mesenteric lymph nodes (mLN), and lamina propria (LP) were excised. LP lymphocytes were isolated using the Miltenyi LP dissociation kit (catalog 130-097-410) as per the manufacturer's instructions. Livers and lungs were digested in a solution of 1 mg/ml collagenase A (Roche) and 75 U DNase I (Sigma-Aldrich) in RPMI 1640 with 5% newborn calf serum. Digested tissues were treated with ACK lysis buffer to remove RBCs and were passed through 100 µm cell strainers. Leukocytes were collected at the interface of a 40%:80% Percoll (Sigma-Aldrich) gradient in RPMI 1640 with 5% NCS. The pelleted cells were washed in 1x DPBS with 2% FBS. Spleens and

mLN were teased apart, treated with ACK lysis buffer, and washed in 1 x DPBS with 2% FBS.

## Fluorescence-assisted cell sorting (FACS) isolation of ILCs from peripheral mouse tissues

Animals were sacrificed, and lymphocytes were isolated from the SI, mLN, and IP lavage as described above. Single-cell suspensions were stained with an e450 Lineage antibody cocktail (Invitrogen, 88-7772-72) and BD Horizon AlexaFluor 700 Fixable Viability Stain (BD 564997). Cells were sorted based on GFP expression on a BD FACSAria II (BD Bioscience), and GFP[+] cells were collected into cR10 prior to downstream processing.

## Patient selection and information

Peripheral blood samples were collected from 12 adults who underwent HSCT at the Duke Adult Bone Marrow Transplant Clinic in Durham, NC between January 2015 and April 2017. All patients signed informed consent under Duke University IRB study protocol Pro00110250. Of the patients who remained stable after allo-HSCT, two out of three received myeloablative conditioning regiments while one underwent non-myeloablative conditioning. Similarly, of the patients who remained went on to experience an episode of acute GVHD in the first four months following their allo-HSCT, two out of three received myeloablative conditioning regiments while one underwent non-myeloablative conditioning. The average patient age at the time of HSCT was 56 years, and all included patients had a transplant indication of myelodysplastic syndrome. Sex and/or gender were not considered in the study design as specimen selection was limited to a small pool of experimentally eligible samples. All patients received calcineurin inhibition and short-course methotrexate for GvHD prophylaxis. We requested samples from 3–6 patients after alloHSCT with a diagnosis of acute graft-versus-host disease and 3–6 that were stable. In addition, we requested a pre-HSCT sample and then one drawn as close as possible to the time of the aGVHD diagnosis, as well as 3 months and 1 year where available.

## ChIP-seq sample preparation

$2.5–5 \times 10^6$ cells were pelleted prior to fixation with and fixed with a 1% formaldehyde using the ChIP-IT High Sensitivity Kit (Active Motif). After quenching and washing, pellets were frozen at $-80\,°C$. Upon thawing, cells were sheared using a chilled dounce homogenizer using with the ChIP-IT High Sensitivity Kit (Active Motif), and lysed cells were sonicated with nanodroplets for 60 s with the Covaris Le220 prior to clarified by centrifuging at full speed for 15 min. Input DNA was prepared with RNase A and Proteinase K prior to clean up and concentration with the Zymo Chip DNA Clean and Concentrate kit. 10–30 ug of ILC chromatin were treated with an anti-H3K4me3 antibody (Cell Signaling Technologies, rabbit mAb 9751, clone C42D8) + blocker mix along with a protease inhibitor cocktail prior to overnight end-to-end rotation at 4 °C. 30 μL of Protein G Dynabeads were added to each 240 μL immunoprecipitation reaction, and reactions were incubated for 3 h. Samples were passed through a ChIP Filtration Column and then washed 5x with Wash Buffer AM1. After the removal of the residual wash buffer, samples were eluted in pre-warmed elution buffer AM4. De-crosslinking was carried out for 2.5 h with Proteinase K prior to DNA extraction with the ChIP DNA Clean & Concentrator (Zymo Research) protocol. Chromatin immunoprecipitation quantitative real-time PCR (ChIP-qPCR) was performed using the QuantStudio 6 K from Applied Biosystems.

## ATAC-sequencing sample preparation

Cells of interest were prepared in suspension and nuclei were isolated by pelleting 25,000–200,000 in a fixed-angle centrifuge. Cells were lysed with 10 mM Tris-HCl (pH 7.4), 10 mM NaCl, and 3 mM $MgCl_2$ prior to 30 min of transposition with an in house Tn5 transposase

(generated at UNC CICBDD with Addgene construct #60240 as adapted from Picelli et al., Genome Res, 2014)[30] at 37 °C. Immediately following transposition, samples were purified with Zymo Conc & Clean (Genesee) and stored at $-20\,°C$ prior to library amplification. Fragments were amplified using 1× NEBnext PCR master mix (New England BioSciences) and custom Nextera PCR primers 1 and 2 (Illumina, Supplementary Data 3). Full libraries were amplified for five cycles (Thermo Fisher), after which a test aliquot of each sample was taken to test 20 cycles to determine the additional number of cycles needed for the remaining 45 μL reaction (QuantStudio 6 K, Applied Biosystems). After the additional cycles were complete (average of 5–15 additional cycles), libraries were purified using a Zymo Conc & Clean (Genesee) prior to a two-sided Ampure bead (Beckman Coulter) size selection to enrich nucleosome-free fragments. Fragment size and concentration were determined by TapeStation 2000 and Qubit 4 (Agilent). Following QAQC, samples were sequenced on the Illumina HiSeq4000 (75x paired-end HO).

## 10x Chromium multiome single-cell library preparation and sequencing

Following FACS sorting, paired multiome Single Cell ATAC (scATAC) and Single Cell Gene Expression (scGEX) libraries were prepared using the 10x Chromium Single Cell Multiome ATAC + Gene Expression kit (CG000338) with the low cell input nuclei isolation protocol (CG000365 Rev C). Lysis buffer strength was 1X, and lysis time was 4 min. Multiome ATAC libraries (50x8x24x9) and GEX libraries (28x8x24x9) were sequenced at a depth of 25–50,000 read pairs per cell on either the Illumina NextSeq2000P2 or Illumina NovaSeq SP.

## mRNA evaluations

The expression of Type 1 and Type 2 lineage-defining and lineage-associated genes was assessed in ex vivo expanded ILC2s and pciILC2s via quantitative real-time polymerase chain reaction (qRT-PCR) was performed with the RT2 Profiler PCR Array system (Qiagen, Hilden, Germany). Briefly, ILC2 and pciILC2 cells were cultured as described above and RNA was extracted after 6 days of expansion with the RNeasy Mini Kit (Qiagen). cDNA synthesis and genomic DNA elimination were performed with the RT² First Strand Kit (Qiagen). cDNA was applied to the RT² Profiler™ PCR Array Mouse Th1 & Th2 Response kit (PAMM-034Z; Qiagen) and PCR amplification was performed using RT2 SYBR Green qPCR Mastermix (Qiagen). Raw CT values were uploaded to Qiagen's web-based software GeneGlobe, where normalization was performed by comparing to the internal housekeeping gene panel, and relative mRNA expression in the ILC2s and pciILC2s was calculated via the ΔΔCT method.

## ChIP-seq analysis

Adapter sequences were removed from reads using cutadapt (v. 1.12). Reads were quality filtered using FASTX-ToolKit (v0.0.12) passing options Q 33, -p 90, and q 20. Reads were aligned to the mm10 genome using STAR (v2.5.2b) with the options: --outFilterScoreMin 1, --outFilterMultimapNmax 1 --outFilterMismatchNmax 2, --chimJunctionOverhangMin 15, --outSAMtype BAM Unsorted, --outFilterType BySJout, --chimSegmentMin 1. Read per million (RPM) normalized H3K4me3 signal at gene (RefSeq) promoters was calculated using deepTools (v.3.5.2), with the promoter region being defined as +300 bp from mm10 RefSeq TSS to −500 bp. Published H3K4me3 fastq files were downloaded from a public repository (GEO: GSE85156)[8] and were aligned to the mm10 genome as described above. H3K4me3 peaks were identified using MACS2 (v.2.1.2, using default parameters)[9] for all subpopulations of ILCs (two replicates each). Differential H3K4me3 marked promoters were identified through the application of DEseq2 (Likelihood ratio test, v.1.40.2)[31] on the union set of peaks. Multiple testing was mitigated using the Benjamini and Hochberg method. ILC1, ILC2, and ILC3-specific H3K4me3 marked promoters

were identified by hierarchical clustering of differential promoters. Representative tracks were created by passing RPM normalized bigwig files to the UCSC Genome Browser.

## RNA-seq analysis

*Data from Bruce* et al. *J Clin Invest, 127(5):1813-1825. doi: 10.1172/JCI91816, PMID: 28375154.* FASTQ files were downloaded from GEO (GSE95811)[4]. Adapter sequences were removed using (cutadapt v1.12). Reads were quality filtered using fastq quality filter in FASTX-Toolkit (v0.0.12) with options -Q 33, -p 90, and q 20. Reads were aligned to mm10 genome using STAR (v2.5.2b) with the following options: --quantMode TranscriptomeSAM, --outFilterMismatchNmax 2, --alignIntronMax 1000000, --alignIntronMin 20, --chimSegmentMin 15, --chimJunctionOverhangMin 15, --outSAMtype BAM Unsorted, --outFilterType BySJout, --outFilterScoreMin 1. Transcript-level expression values (TPM) were estimated using Salmon (v0.11.3) for RefSeq transcripts. TPM data was collapsed into gene-level expression values using a tximport. Refseq was used for gene annotation.

## ATAC-seq analysis

Sequence adapters were trimmed using cutadapt (v. 1.12) with the following options -a CTGTCTCTTATA -A CTGTCTCTTATA and --minimum-length 36 and set for paired-end sequence data. Reads were then quality filtered using fastq_quality_filter in FASTX-Toolkit (v. 0.0.14), with the options -Q 33, -p 90, and -q 20. PCR duplicates were restricted by limiting the same sequence to a max of 5 copies. After this duplicate threshold was reached, the remaining reads were excluded (In house scripts). As one read of a read-pair may have been excluded during filtering, fastq files were synced following the filtering step to ensure accurate alignment. Reads were aligned to either mm10 or hg38 using STAR (v2.5.2b) with options –chimSegmentMin 15, --outFilterMismatchNmax 2, --chimJunctionOverhangMin 15, --outSAMtype BAM Unsorted, --outFilterScoreMin 1, --outFilterType BySJout and --outFilterMultimapNmax 1. Samtools (v. 1.3.1) and bedtools (v. 2.26) were then used to create bigwig files. These files only include fragment data for the first 5 bp from the 5' end and the last 4 bp from the 3' end of fragments. Regions of chromatin accessibility (peaks) were defined using MACS (v. 2.1.2) with options –nomodel, --shift 0, --extsize 5. Sites of differential chromatin accessibility were identified using DESeq2 (v. 1.40.2) with default parameters using a union set of peaks. Motif analysis was performed using findMotifsGenome.pl from HOMER (v. 4.11.1). Representative tracks of normalized ATAC signal were generated by visualizing bigwig files on the UCSC genome browser.

## MARS-seq data

*From Gury-BenAri* et al. *Cell, 2016, 166(5):1231-1246.e13, doi: 10.1016/j.cell.2016.07.043, PMID: 27545347.* The UMI count table was downloaded from GEO (GSE85152). The count table was calculated as previously described[8]. The first gene name was selected to represent features when more than one possible annotation was available. Cells with less than 200 UMIs were discarded from downstream analysis.

## single nucleus RNA and ATAC-seq analysis

Paired-end FASTQ files were pre-processed and aligned to the mouse reference genome (GRCm38/mm10) using cellranger-arc count (v. 2.0.0) with default parameters. Next, we selected nuclei with a total RNA read count < 25,000, total RNA read count >1000, total ATAC read count <70,000, total ATAC read count >5000, and mitochondrial counts < 20% for downstream analysis. We then integrated the datasets using RNA data only or with both RNA and ATAC. Integration using RNA was accomplished using the IntegrateData function within Seurat passing default parameters (v. 4.3.0.1)[10]. The dimensionality of the data was reduced using principal component analysis (PCA), and Uniform Manifold Approximation and Projection for Dimension Reduction (UMAP). Cells were then clustered using the FindClusters

function within Seurat. Prior to the integration of multi-modality data, the RNA and ATAC count matrices for each sample were merged. The merge function within Seurat was used to concatenate the RNA count matrices. To create the merged ATAC count matrix, we first identified a union set of peaks across all samples and calculated the number of ATAC fragments overlapping those shared genomic coordinates for each sample using FeatureMatrix (Signac v. 1.10.0). The RNA and ATAC count matrices were then merged by sample, and the data were integrated using the Weighted nearest neighbor analysis (Seurat) to develop shared inference[32]. Differentially expressed genes were identified using a Wilcoxon rank sum test as implemented in the package presto (v. 1.0.0) or with the FindMarkers function in Seurat. Heatmap of differentially expressed genes depicts mean-centered and normalized RNA abundance. Normalized (SCTransform) and uncorrected (pre-integration) RNA counts were used for differential testing. A 2-cell cluster (pre- and post-transplant mouse data, AAGGATTAGCTCATAA-1_2, AGTGGACAGCTATTAG-1_2) was identified and excluded from downstream analysis. To determine differential regions of chromatin accessibility, we also used a Wilcoxon rank sum test (presto, v. 1.0.0) on TF-IDF normalized ATAC counts. Heatmap of differential regions of chromatin accessibility depicts mean-centered and normalized ATAC signal. Per-cell accessibility for motifs (motif enrichment scores) was calculated using the package chromVAR (v. 1.22.1). The JASPER 2020, CORE, vertebrate motif database was used during motif identification. To identify putative TF regulators for each cluster of cells, the AUC statistics calculated by presto during the differential expression and differential motif enrichment analysis (Wilcox rank sum test (p adj < 0.05, RNA.logFC > 0 and motif.p adj < 0.05, motif.logFC > 0)) were averaged, and the TFs with the highest value were considered top candidates. Average per-cell chromatin accessibility (ATAC signal) across genes previously associated with ILC1 and ILC2 isolated from small intestinal lamina propria of mice[8] was calculated using the FeatureMatrix from the Signac Package. For ontology analysis, we used Enrichr[33] and tested for significant overlap between genes associated with post-transplant cells and gene sets found in the mouse gene atlas database. The average marker expression of ILC1, ILC2, ILC3, and NK cells was defined by averaging the normalized gene expression for genes previously associated with each cell type[11]. Classification of post-transplant cells as ILC1, ILC2, or ILC3 was based on lineage defining genes previously associated with each type of ILC[8]. Seurat was used to integrate post-transplant single nucleus RNA data with previously published[8] MARS-seq RNA abundance data as previously described[10]. Cells were classified by calculating the average rank of genes associated with each type of ILC and labeling each cell with the cell type with the highest rank. "Undetermined" signifies a tie in ranking. Upset plots were created using UpSetR (v. 1.4.0).

## Single-cell ATAC analysis

Paired-end FASTQ files were pre-processed and aligned to the mouse (GRCm38/mm10) or human reference genome (GRCh38/hg38) using cellranger-atac count (v. 2.0.0) with default parameters. To compare the average normalized ATAC signal across each patient sample at hILC2 sites of CA (defied with bulk ATAC data), we first calculated the number of ATAC fragments overlapping the 7456 associated sites of CA using FeatureMatrix (Seurat, v. 4.3.0.1), respectively. We then merged these count matrices and normalized the data using TF-IDF as implemented in Seurat (v. 4.3.0.1). The same operations were performed for the pc-hILC2 associated sites of CA ($n = 5,051$) as well as random sites ($n = 7,456$) of chromatin accessibility shared between hILC2 and pc-hILC2s. Samples were grouped into six patient conditions, four according to transplant status and aGVHD, two healthy donors, and expanded ILC2s derived from an additional single healthy donor. Differential ATAC was tested using patient condition as a categorical variable in a linear model. The model was fit in R and the coefficients were evaluated to highlight different effects for patient

 

conditions. We then grouped the sample into four categories: healthy donor, ILC2, pre- and post-transplant samples (independent of aGVHD) and repeated the above analysis. For both tests, we assessed significance using $p < 0.05$. Identification of genes with differential ATAC signals between pre- and post-transplant patients was performed using the sum of ATAC signals at each gene for all patient samples (calculated with FeatureMatrix). The 6 pre-transplant and 6 post-transplant samples were then analyzed as biological replicates of each condition, and differential analysis was performed using DESeq2. Ontology analyses were performed using g:profiler (v. 0.2.2)[34]. For differential analysis of aggregate ATAC signal for each gene average ATAC signal was defined as the total normalized ATAC signal over all gene bodies plus 2 kb upstream for all pre- ($n = 6$) and post-transplant ($n = 6$) patient samples. Genes with differential ATAC signals were then identified using DESeq2.

### Statistical analysis

Survival differences were evaluated using a Mantel-Cox log-rank test. Survival curves were generated using the Kaplan-Meier method. Differences in GVHD clinical and pathology scores were determined using 2-way ANOVA, with Bonferroni correction for repeated measures of multiple comparisons. Statistical analysis of ATAC-seq and RNA-seq data is described above. Unless otherwise noted in the figure legends, all other continuous variables were compared using a 2-tailed Student's $t$ test with Welch's correction. A $P$-value of $\leq 0.05$ was considered statistically significant.

### Reporting summary

Further information on research design is available in the Nature Portfolio Reporting Summary linked to this article.

## Data availability

The data generated in this study have been uploaded to the Gene Expression Omnibus (GEO) database where they are stored under the accession number GSE232003. RNA-seq data from Fig. 1b were downloaded from GEO using the accession number GSE95811. H3K4me3 iChIP-IVT and MARS-seq data were downloaded from GEO accession numbers GSE85156 and GSE85152, respectively. Source data are provided with this paper.

## Code availability

The code used to perform analyses and generate figures has been archived in the Zenodo repository (https://doi.org/10.5281/zenodo.11396929) and is available on GitHub (https://github.com/j-foster2/GvHD/releases/tag/v1.0.1).

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

## Acknowledgements

S.J.L. discloses support for the research of this work from the NIGMS (1K12GM000678-20) and NCI (1T32CA211056-01 A1). J.P.F. discloses support for the research of this work from NIGMS (R01GM138912 and T32GM067553). I.J.D. is the G. Denman Hammond Professor for Childhood Cancer at UNC-Chapel Hill School of Medicine and discloses support for the research of this work by NIH grants (R01GM138912 and R01CA276663) as well as grants from the V Foundation for Cancer Research. J.S.S. is the Elizabeth Thomas Professor of Medicine at UNC-CH SOM and is supported by the NIH (grants 1R01HL139730-03 and 1R01HL155098-01A1). We thank Janet Dow at the UNC Flow Cytometry Core for cell sorting assistance and guidance. The UNC Flow Cytometry Core Facility is supported in part by the P30 CA016086 Cancer Center Core Support Grant to the UNC Lineberger Comprehensive Cancer Center. Research reported in this publication was supported in part by the Center for AIDS Research award number 5P30AI050410 and the North Carolina Biotech Center Institutional Support Grant 2012-IDG-1006. We gratefully acknowledge the technical support from the UNC High Throughput Sequencing Facility. This facility is supported by the University Cancer Research Fund, Comprehensive Cancer Center Core Support grant (P30-CA016086), and UNC Center for Mental Health and Susceptibility grant (P30-ES010126). This work was also supported by the UNC Advanced Analytics Core (Center for GI Biology and Disease; NIH funding number P30 DK034987). The content is solely the responsibility of the authors and does not necessarily represent the official views of the National Institutes of Health. Figures 1a, 2a, 3a, 4a, and e were created with BioRender.com and released under a Creative Commons Attribution-NonCommercial-NoDerivs 4.0 International license.

## Author contributions

S.J.L. and J.S.S. conceived the project. S.J.L., J.P.F., J.S.S., and I.J.D. devised the methodology. S.J.L., D.W.B., O.V.K., M.Y., and H.B. performed experiments. S.J.L., J.P.F., J.S.S., and I.J.D. wrote the original manuscript. S.J.L., J.P.F., J.S.S., I.J.D., S.G.P., J.P., N.J.C., and S.S. edited the manuscript. S.J.L. and J.P.F. analyzed, visualized, and interpreted data. J.S.S., S.J.L., and I.J.D. acquired funding. S.G.P., J.P., S.S., and N.J.C. provided essential tools and insights.

## Competing interests

J.S.S. has received research funding from Merck Inc., Carisma Therapeutics, and Glaxo Smith Kline and is a compensated consultant for PIQUE Therapeutics. J.S.S./D.B. Intellectual Property: 16,598,914, D.B./J.S.S.: US patent 11,471,517. S.G.P. and I.J.D. own equity in Triangle Biotechnology, Inc. S.G.P. is an inventor of Intellectual Property related to this research that is licensed to Triangle Biotechnology, Inc. from the University of North Carolina at Chapel Hill. The remaining authors declare no competing interests.

## Additional information

[1]Lineberger Comprehensive Cancer Center, University of North Carolina School of Medicine, Chapel Hill, NC, USA. [2]Curriculum in Bioinformatics & Computational Biology, University of North Carolina, Chapel Hill, NC, USA. [3]Center for Integrative Chemical Biology and Drug Discovery, Division of Chemical Biology and Medicinal Chemistry, University of North Carolina Eshelman School of Pharmacy, Chapel Hill, NC, USA. [4]Division of Hematologic Malignancies and Cellular Therapy, Department of Medicine, Duke University Medical Center, Duke Cancer Institute, Durham, NC, USA. [5]Department of Genetics, University of North Carolina, Chapel Hill, NC, USA. [6]Division of Pediatric Hematology-Oncology, Department of Pediatrics, University of North Carolina School of Medicine, Chapel Hill, NC, USA. [7]Department of Microbiology & Immunology, University of North Carolina School of Medicine, Chapel Hill, NC, USA. [8]Division of Hematology, University of North Carolina School of Medicine, Chapel Hill, NC, USA. [9]Department of Medicine, University of North Carolina School of Medicine, Chapel Hill, NC, USA. [10]Present address: Duke Eye Center, Duke University, Durham, NC, USA. [11]These authors contributed equally: Sonia J. Laurie, Joseph P. Foster II. ✉e-mail: jonathan_serody@med.unc.edu

