## [Peer Review File · Nature Communications]

Type II innate lymphoid cell plasticity contributes to impaired reconstitution after allogeneic hematopoietic stem cell transplantationREVIEWER COMMENTS

Reviewer #1 (Remarks to the authors)

Laurie and colleagues performed a study on ILC2 in a mouse model of acute GvHD. They investigated the fate of transplanted ILC2 in allo-HSCT recipients, and demonstrated that after transplantation, the transferred (GFP+) ILC population is dominated by NK and ILC1-like cells. They postulate that ILC2 reconstitution after allo-HSCT is hampered by the plasticity of ILC2. This work is important as ILC have been demonstrated to have protective effects against acute GvHD, both in mice and in men. The study raises however also quite a few points:

1. The main purpose of the study is to '... evaluate the hypothesis that the absence of ILC2 post-transplant is mediated in part by conversion of these cells to an alternate fate.' (lines 49-51). The authors demonstrate indeed that small intestine ILC2 transition to NK and ILC1-like cells after transplantation, but whether this accounts for delayed ILC2 recovery remains to be answered, in a large part because actual ILC reconstitution data (including NK cells) are lacking. How are NK cell, ILC1 and ILC1-like cell recovery dynamics post allo-HSCT in this model and does NK/ILC1/ILC1-like recovery account for (part of) the 'missing' ILC2 after allo-HSCT?

2. In the past, the authors have demonstrated in mice that adoptive transfer of ILC2 delayed the development of acute GvHD (Bruce et al, JCI 2017 and Blood Adv 2021). Here they demonstrate that part of transferred ILC2 are plastic and transition into a NK cells and an ILC1-like population, at least in the small intestine. It remains unclear A) how large these populations are, B) whether the extent of plasticity or number of cells that convert into NK and ILC1-like cells is related to the extent of acute GvHD that these mice develop and C) how these data relate to the beneficial effect of adoptive transfer of ILC2. The fact that this plasticity is observed in all mice suggests that this transition is a bystander effect and not 'clinically' relevant.

3. It turned out difficult to push ILC2 towards the NK and ILC1-like status *ex vivo*. In these experiments a pro-inflammatory cytokine cocktail was used. Have the authors tried other stimuli relevant for the post-transplant setting, such as DAMPs (extracellular ATP, others), microbes or microbial products? Along the same lines: does transformation of ILC2 to NK and ILC1 like cells also happen in lethally irradiated mice transplanted with BM+ILC2 (no T cells)? In other words, could tissue damage/bacterial translocation itself play a role? Or perhaps homeostatic proliferation? What happens when ILC2 are transferred to antibody-depleted mice (where there is no tissue damage)?

4. To demonstrate relevance of exILC2/ILC1 like cells the authors transferred the *ex vivo* cultured ILC2 into irradiated mice, together with BM and T cells, and found that these pcILC2s (pc: pro-inflammatory conditioned) exacerbated acute GvHD. By stating in the Abstract that 'Exposure of ILC2s to proinflammatory cytokines *in vitro* resulted in a mixed ILC1-ILC2 phenotype but was able to convert only an extremely small population of ILC2s to ILC1s as observed *in vivo*. [...] infusion of proinflammatory cytokine-exposed ILC2s accelerated aGvHD' it is suggested as if the ex-ILC2/ILC1 like cells found *in vivo* contributed to inflammation and GvHD. In lines 203-214 a similar link is suggested between the plasticity of transferred ILC2, the ex-ILC2/ILC1 like cells, pcILC2 and the human situation. The majority of transferred pcILC2 are however clearly different from the ex-ILC2/ILC1 like population that is found after adoptive transfer and in humans this ex-ILC2/ILC1 like population was not even demonstrated. The plasticity of ILC2 when transferred with the BM+T cell transplant in mice is a different story and there are no convincing data that these can be linked to the pcILC2. Also the title of the manuscript is an overinterpretation of the data: the presented data do not prove that '.... inflammatory cytokines affect ILC fate after allo-HSCT'.

5. Also the connection between the mouse and human data is weak and suggestive. Based on PBMC of 3 patients with and 3 without acute GvHD the authors suggest similar pathophysiologic mechanisms between mouse GvHD models and human acute GvHD. Aside from the fundamental differences between murine GvHD models and human acute GvHD, fundamental differences have been noted between mouse and human ILC. For example, ILC2 are absent in the human GI tract. I do not see how the human data presented here that are obtained from unsorted peripheral blood PBMC can in any way be related to the findings from GI-tract isolated ILC from mice. In healthy

circumstances ILC comprise <0.5% of PBMC, and this is significantly less in human allo-HSCT recipients 4-6 weeks after transplant, in particular when these patients have acute GvHD. Basically, the human data only demonstrate that patients without acute GvHD have higher ILC2 numbers in the blood which has been demonstrated before.

In addition:

6. The authors should make much more clear in the title, abstract and throughout the manuscript that the findings are related to a mouse model of acute GvHD.

7. Fundamental characteristics of the patients under study are missing, such as conditioning regimen (myeloablative or RIC), properties of the transplant (lymphocyte depleted or lymphocyte-replete), GvHD prevention regimen and immunosuppressants used at the time of blood draw, and time since transplantation. Without these data it is challenging to draw conclusions from the human data.

8. Figure 1A: what is the purpose of treating mice with IL-25? Please explain. If it is to enhance the yield of ILC2, then please elaborate on how well these cells represent true, unstimulated ILC2. Figure 1C: how were cells purified? Based on which parameters? Are ILC1 derived from IL25 stimulated mice bona fide ILC1? Figure 1E: cells gated as lineage neg/ ST2 pos? or show ST2 expression? ST2 is mentioned in legend but not shown in graphs. Extended data Figure 1C-E: please clarify R1 and R2.

9. Figure 2C: an alternative explanation to plasticity of ILC2 could be that the 0.3% of ILC in the pre-transplant population have proliferated and seeded the post-transplant population. Assuming exponential growth, only few rounds of division are needed. How was this excluded?

10. Figure 3L, M: Survival advantage of mice that received ILC2 not significant in contrast to previous publications (JCI 2017, Blood Adv 2021). Please clarify. In the text (lines 167-168) it is stated that '... co-administration of ILC2 significantly diminished clinical GvHD scores and improved survival...' but this is not supported by the data.

11. In lines 215-222 it is concluded that NR4A2 and FLI1 are novel regulators of ILC2 plasticity. However, functional experiments eg. knockout experiments to demonstrate a functional relation to back this statement are lacking. Also it is unclear whether the authors refer to mouse or human ILC here.

Reviewer#2 (Remarks to the authors)

The paper by Laurie et al builds on observations in mice and humans that ILC2 fail to reconstitute after allo-HSCT. Work in a mouse model indicate that this is caused by conversion of ILC2 to ILC1-like cells. Data are presented that suggest a similar mechanism prevents ILC2 reconstitution in humans as well.

That ILC2 fail to reconstitute in allo-HSCT in humans is, however, incorrect. Ref 6 describe that ILC2 do reconstitute but at a slower pace than the other ILC subsets. There is no evidence in humans that ILC2 protect against acute GVHD. Ref6 seems to indicate that ILC3 are mediating this effect.

Whereas the data in the mouse model are generally supporting the interpretation of the authors. However, it is difficult to judge whether the pcILC2 are really derived from the ILC2 and not from ILC1s contaminating the ILC2 population and expanded in the ILC1 permissive condition. The authors need to provide information of the purity of the starting cell population and the expanded cells.

it is not possible to judge the human data. No information is given of how the human ILC2s are isolated and what the phenotype is of these cells before and after culture in ILC2 or ILC1 permissive conditions.

Reviewer #3 (Remarks to the Author):

I center my review around the reproducibility of data analysis. Following a meticulous assessment of the GitHub repository, I can confirm that all the issues related to data analysis reproducibility have been successfully addressed.

As a major recommendation, I suggest including a section in the manuscript dedicated to data availability. This section should specify the GEO repositories, where the generated data can be accessed.

Additionally, as a minor suggestion, I recommend expanding the GitHub repository's readme file. Specifically, consider adding an introductory chapter that succinctly summarizes the purpose of the GitHub repository and its connection to the associated research paper.

22 March 2024

We greatly appreciate the insightful and thorough review of our manuscript which has clearly strengthened our manuscript. We believe the reviewers found merit in the submission but raised multiple important points for clarification. Below, we will detail our point-by-point response to the previous review.

Laurie et al responses to Reviewer #1

Reviewer 1, Remark 1. *“The authors demonstrate indeed that small intestine ILC2 transition to NK and ILC1-like cells after transplantation, but whether this accounts for delayed ILC2 recovery remains to be answered, in a large part because actual ILC reconstitution data (including NK cells) are lacking. How are NK cell, ILC1 and ILC1-like cell recovery dynamics post allo-HSCT in this model and does NK/ILC1/ILC1-like recovery account for (part of) the ‘missing’ ILC2 after allo-HSCT?”*

As the reconstitution of NK cells has been well documented in previous manuscripts [Foley et al. (*Blood*, 2011, PMID: 21757615), Inamura et al. (*Hematology*, 2003, PMID: 12623423), and van den Brink (*Bone Marrow Trans*, 2018, PMID: 29367714)], we did not pursue studies evaluating NK cell reconstitution in our standard haploidentical model. However, to begin to address the question regarding ILC cells, we evaluated the number of Innate lymphoid type 1 (ILC)1 cells in the liver post allogeneic hematopoietic stem cell transplantation (allo-HSCT). ILC1 cells were reduced, although not to the extent of ILC2 cells, suggesting that ILC generation may be globally affected post allo-HSCT. Although this is an interesting finding, the recovery of ILC1 cells is outside the scope of this manuscript, which focuses on ILC2 cells. Future work will evaluate mechanisms underlying the diminished recovery of other ILC subsets post HSCT.

Reviewer 1, Remark 2. *“It remains unclear A) how large these populations are, B) whether the extent of plasticity or number of cells that convert into NK and ILC1-like cells is related to the extent of acute GvHD that these mice develop and C) how these data relate to the beneficial effect of adoptive transfer of ILC2. The fact that this plasticity is observed in all mice suggests that this transition is a bystander effect and not ‘clinically’ relevant.”*

We thank the reviewers for providing us with an opportunity to better define our description of the population of cells recovered from the small intestine (SI) following allo-HSCT. ILC2 cells, while constituting a small population of total innate cells in the GI tract, are found in the intraepithelial compartment (IEC) of the colon and SI in human evaluations (Yudanin et al, *Immunity*, 2019, PMID: 30770247). The current studies were designed to provide

insight into the lack of ILC2 cells in the IEC compartment of the small bowel after allo-HSCT despite their presence in that compartment in healthy individuals and their circulation in the bloodstream in healthy individuals and those with inflammatory diseases of the GI tract. These studies were not designed to correlate the effect of donor ILC2 cells on alloreactive T cell expansion or how their function affects donor T cell activity after allo-HSCT. Here, we show that the infusion of ILC2 cells leads to the presence of ILC1-like cells in the SI providing the first evidence that the absence of ILC2 cells after allo-HSCT may be due to inflammation-driven conversion of ILC2 cells. That almost all the ILC2 cells convert to ILC1-like cells is not a “bystander effect” (term used in the review) but may be the cause of their marked reduction in the GI tract in GvHD. To support this hypothesis, the number of donor ILC2 cells that can be recovered from the GI tract in allogeneic recipients is much less than those recovered from syngeneic recipients. Indeed, we found that the number of recovered marked donor ILC2 cells was quite modest, and as a result required the pooling of samples for many of our analyses. While we have not attempted to perform transplants titrating in the number of pcILCs and thus do not currently have data enabling us to correlate the extent of ILC2 conversion to the severity of aGvHD, as we document in **Panels L and M of Figure 3**, infusion of between 3-4 x10⁶ pro-inflammatory conditioned ILC2s (pcILC2s) failed to mitigate aGvHD in this model and instead was associated with an increase in disease-mediated morbidity and mortality.

Reviewer 1, Remark 3, Part A. *“It turned out difficult to push ILC2 towards the NK and ILC1-like status ex vivo. In these experiments a pro-inflammatory cytokine cocktail was used. Have the authors tried other stimuli relevant for the post-transplant setting, such as DAMPs (extracellular ATP, others), microbes or microbial products?”*

To augment the in vitro transdifferentiation of ILC2s to more completely achieve the type of conversion noted in vivo following murine HSCT, we undertook a series of experiments in which we added other stimuli relevant for the post-transplant setting. We treated ILC2 cells with resiquimod, a TLR 7/8 agonist, or HMGB-1, a nonhistone chromatin-associated protein, which when released by dying cells serves as a robust damage-associated molecular pattern molecule (DAMP) and whose presence has been correlated with GvHD (Apostolova & Zeiser, *Human Immunology*, 77(11), 2016, PMID: 26902992). The addition of either these immune-stimulatory microbial- or damage-associated products on either Day 2, Day 4, or Days 2 and Day 4 during pro-inflammatory cytokine-mediated skewing did not result in substantial changes to cell viability. Additionally, it failed to lead to the generation of more Tbet-expressing or IFN- γ -secreting cells. New text describing these findings has been inserted and highlighted on **lines 181-187** and data have been inserted as new panels in **Extended Data Figure 4d, e**.

Reviewer 1, Remark 3, Part B. *“Does transformation of ILC2 to NK and ILC1 like cells also happen in lethally irradiated mice transplanted with BM+ILC2 (no T cells)? In other words, could tissue damage/bacterial translocation itself play a role? Or perhaps homeostatic*

proliferation? What happens when ILC2 are transferred to antibody-depleted mice (where there is no tissue damage)?”

To address these questions, we performed two different bone marrow transplant evaluations. In the first transplant model, we infused B6 donor TdTomato+ ILC2 cells with B6 donor TCD bone marrow cells **without** donor T cells. We then evaluated for the presence of TdTomato+ ILC2 cells on day 21 post-transplant. This model evaluates the expansion of ILC2 cells in an allogeneic setting but without GvHD-inducing T cells. There was a significant decrease in the recovery of donor ILC2 cells in the SI in this setting compared to mice receiving donor T cells suggesting that donor T cells may be needed for donor ILC2 migration or expansion. Thus, we could not evaluate for conversion in this setting. For the second proposed experiment, we transplanted syngeneic B6 ILC2-TdTomato cells with B6 T cells and B6 TCD BM cells into lethally irradiated B6 mice and isolated TdTomato+ cells in the SI on day 21 post-transplant. In this experiment, we detected a large population of donor ILC2 cells, which was significantly greater than that found in the allogeneic setting. These data support a model in which donor T cells enhance the presence of donor ILC2 cells in the SI, which is significantly decreased in allogeneic recipients compared to syngeneic recipients. These data also suggest that the alloreactive immune response may limit the recovery of ILC2 cells in the GI tract supporting our hypothesis that inflammation is important for the limited presence of these cells in the GI tract. These data have been added to the current version of the manuscript.

Reviewer 1, Remark 4. *“To demonstrate relevance of exILC2/ILC1 like cells the authors transferred the ex vivo cultured ILC2 into irradiated mice, together with BM and T cells, and found that these pcILC2s (pc: pro-inflammatory conditioned) exacerbated acute GvHD. By stating in the Abstract that ‘Exposure of ILC2s to proinflammatory cytokines in vitro resulted in a mixed ILC1-ILC2 phenotype but was able to convert only an extremely small population of ILC2s to ILC1s as observed in vivo. [...] infusion of proinflammatory cytokine-exposed ILC2s accelerated aGvHD’ it is suggested as if the ex-ILC2/ILC1 like cells found in vivo contributed to inflammation and GvHD. In lines 203-214 a similar link is suggested between the plasticity of transferred ILC2, the ex-ILC2/ILC1 like cells, pcILC2 and the human situation. The majority of transferred pcILC2 are however clearly different from the ex-ILC2/ILC1 like population that is found after adoptive transfer and in humans this ex-ILC2/ILC1 like population was not even demonstrated. The plasticity of ILC2 when transferred with the BM+T cell transplant in mice is a different story and there are no convincing data that these can be linked to the pcILC2. Also, the title of the manuscript is an overinterpretation of the data: the presented data do not prove that ‘... inflammatory cytokines affect ILC fate after allo-HSCT’.”*

We thank Reviewer 2 for taking the time to detail these concerns. We agree that the majority of our in vitro-derived pcILC2 are transcriptionally distinct from the ILC1-like ex-ILC2 identified in the SI after adoptive transfer and subsequent HSCT. However, although not identical to ILC1s, we found that exposure to skewing cytokines results in transcriptional and epigenetic changes associated with a type 1-like fate, with limited expression of type 1 defining

TFs. Despite this, quantification of ILC associated genes by qPCR array identified increased expression of *Tbx21*, *Stat1* and *Ifng* in pcILC2s. Additionally, differential gene expression analysis of transcripts detected by snRNA-seq revealed a significant increase in ILC1-associated *Stat1* and *Bcl2* in pcILC2s, and decreased expression of ILC2 associated *Gata3*. Cytokine treatment also resulted in a significant increase of chromatin accessibility proximal to *Tbx21* and *Ifng* (Figure 3F). To further explore our hypothesis that pcILC2s represent an intermediate/transition state between ILC2s and ILC1-like cells, we compared gene expression in the pcILC2 cells to a set of genes that are expressed by both ILC1 and ILC2 cells, rather than either cell type (Figure 1B, Gury-BenAri et al, *Cell*, 2016, PMID: 27545347). We found the expression of these genes is significantly increased in the cytokine treated cells compared to untreated ILC2s. In these cells, the greatest decrease was found in ILC2 associated genes, as well as reduced expression of genes that are expressed in both ILC2 and ILC3 cells. Taken together, these data suggest that proinflammatory cytokine treated ILC2s may be in an intermediate state, transitioning from an ILC2 to a combined ILC2/ILC1-like cell state. Text for the new analysis can be found at **lines 172-178** and the corresponding panel has been added to **Figure 3i**. Unfortunately, lacking access to multiome data from the GI tract of patients undergoing allo-HSCT that develop GvHD, we cannot evaluate for the presence of plasticity of ILC2 cells in patients after allo-HSCT. However, we have shown that there is a decrease in signal at sites of accessible chromatin specific to ILC2 cells in circulating immune cells after allo-HSCT, which is like that found in our cytokine-exposed ILC2 cells. Finally, we agree with the reviewer regarding the title and have altered the language to more precisely reflect the findings of our manuscript.

Reviewer 1, Remark 5. *“The connection between the mouse and human data is weak and suggestive. Based on PBMC of 3 patients with and 3 without acute GvHD the authors suggest similar pathophysiologic mechanisms between mouse GvHD models and human acute GvHD. Aside from the fundamental differences between murine GvHD models and human acute GvHD, fundamental differences have been noted between mouse and human ILC. For example, ILC2 are absent in the human GI tract. I do not see how the human data presented here that are obtained from unsorted peripheral blood PBMC can in any way be related to the findings from GI-tract isolated ILC from mice. In healthy circumstances ILC comprise <0.5% of PBMC, and this is significantly less in human allo-HSCT recipients 4-6 weeks after transplant, in particular when these patients have acute GvHD. Basically, the human data only demonstrate that patients without acute GvHD have higher ILC2 numbers in the blood which has been demonstrated before.”*

We thank the reviewers for the opportunity to clarify the connection between the mouse and human data. First, ILC2 cells can be found in the GI tract in healthy humans (Yudanin et al, *Immunity*, 2019) by spatial analysis and by single cell analysis (Jarosch et al, *Cell Rep Med*, 2023, PMID: 37467715). Second, we are not attempting to replicate our murine findings using the human samples, which would require that we perform multiome analysis from the GI tract in patients before and after the infusion of donor ILC2 cells, which is not currently possible.

Despite this limitation, we were interested in evaluating if the inflammatory environment post allo-HSCT in patients would lead to chromatin alterations like that found in the tissue of mice with GvHD. This is an important evaluation as ILC plasticity could also occur in the bloodstream where significant quantities of pro-inflammatory cytokines are found post-HSCT.

To test this hypothesis, we performed scATAC-seq on unsorted PBMCs from healthy donors, and patients with and without aGVHD following allo-HSCT and evaluated the average ATAC signal at sites of chromatin accessibility that we had associated with ILC2s. To perform this evaluation, we first identified a set of chromatin regions associated with ILC2s by performing ATAC-seq on ILC2s and cytokine-skewed ILC2s (pc-hILC2s) and identifying ILC2-specific regions of chromatin accessibility relative to pc-hILC2s. PBMCs collected from patients who developed aGVHD following allo-HSCT had a significant reduction in chromatin accessibility signal at ILC2-associated genomic regions. Perhaps as interesting, the signature for pc-hILC2 cells was increased after allo-HSCT regardless of aGVHD status suggesting that the pro-inflammatory environment after allo-HSCT is associated with changes in chromatin state independent of acute GvHD. Furthermore, we also analyzed these data after selecting cells without detectable RNA for MPO, CD3, CD19, CD20 and CD22 to exclude myeloid, T and B cells. The relative decrease in ILC2-associated chromatin signal was also detected in this filtered cell set (**See Figure Below**). We recognize that we are unable to identify precisely the cells that contribute to the signal detected at the ILC2-associated regions. However, as Th2 cells aren't present in the bloodstream after allo-HSCT, we believe that the ILC2 ATAC signal is likely to originate from progenitors, and emerging innate lymphoid cells. Recognizing the challenges in performing these experiments, we consider these interesting and important findings. Future experiments to address our hypothesis regarding the role of inflammation in the conversion of these cells in the bloodstream would require a single cell analysis of chromatin accessibility in patients receiving reduced intensity conditioning or antibody-only conditioning.

To further address the reviewer's concern, we also sought additional evidence that the ILC2-ILC1-like conversion occurs in the human lower GI tract. A study investigating the composition of immune cells in the human GI tract following allo-HSCT was published last year (2023) in *Cell Reports Medicine* by Jarosch and colleagues in Dirk Busch's group. In this study, the authors collected 31 punch biopsies from multiple sites in the human gut following radiotherapy and allo-HSCT from 26 patients. scRNA-seq was performed on each of these samples, and cells were annotated with cell type using cell-surface proteins serving as ILC lineage markers. Using these data, Jarosch et al. quantified the immune cells, including the fraction of innate lymphoid cells. Although the authors assessed the change in ILC cell count across multiple tissues, we focused on their colon data. We noted there were more ILC2 cells in the control samples than in biopsies from patients with aGVHD (Figure 2A, Jarosch et al., *Cell Rep. Med*, 2023). However, a power analysis based on the effect size presented in Figure S2A confirmed that the number of available samples was insufficient to detect a statistical difference.

Finally, we would refer the reviewers to a significant number of manuscripts that detail the presence of ILC2 cells in multiple intestinal compartments in humans (Goc et al, *Cell*, 2021,

PMID: 34407392, Krämer et al, *PLoS Path*, 2017, PMID: 28505204, Moller et al, *Mucosal Immunol*, 2023, PMID: 37121384). Further, while human ILC2 RNA signatures have been detected in both colon and gastric cancers (Huang et al, *Cancers*, 2021, PMID: 33535624, O'Keefe et al, *Nature Communications*, 2023, PMID: 37898600), our study is the first that we that we have identified that evaluates ILC2 signatures in human PBMCs at the chromatin level. Taken together, these published data show ILC2s are in the human lower GI tract and that GvHD is associated with changes in chromatin state in PBMC for ILC2-lineage defining genes. Whether the absence of these cells is associated with the severity of acute GvHD will require analysis of a much larger subset of patients.

Figure 1: ATAC-seq evaluation of PBMC from patients or healthy controls after filtering out cells expressing *Cd19*, *Cd20*, *Cd22*, *Mpo*, *Cd3e*, *Cd3g* and *Cd3d*.

throughout the manuscript that the findings are related to a mouse model of acute GvHD.”

We appreciate the reviewers' concern regarding the title of the manuscript and have modified the title to “Multiomic Evaluation Reveals ILC2 Cell Plasticity After Allogeneic Stem Cell Transplantation”. Additionally, we have added the term murine in multiple areas of the

current version to provide clarity to the reader when the analysis is performed using mouse or human tissues.

Reviewer 1, Remark 7. “Fundamental characteristics of the patients under study are missing, such as conditioning regimen (myeloablative or RIC), properties of the transplant (lymphocyte depleted or lymphocyte-replete), GvHD prevention regimen and immunosuppressants used at the time of blood draw, and time since transplantation. Without these data it is challenging to draw conclusions from the human data.”

We apologize for neglecting to include these data in our initial submission. A description of the study patients has now been added in lines **572-582**.

Reviewer 1, Remark 8, Part A. *Figure 1A: what is the purpose of treating mice with IL-25? Please explain. If it is to enhance the yield of ILC2, then please elaborate on how well these cells represent true, unstimulated ILC2.*

We acknowledge that in the original manuscript the process by which we isolated murine ILC2s in this model in **Figure 1A** was unclear and incomplete. IL-25 was administered intraperitoneally for four days prior to the isolation of ILC2s in order to enhance their accumulation, as they represent a very small fraction of peritoneal cells at steady state (Huang et al, *J Am Soc Neph*, 2015, PMID: 25556172). We recognize that IL-33 can also be used to elicit and activate ILC2s. However, IL33 is more relevant in the airway, endometrium, and tumor microenvironments rather than the GI tract (Li et al, *Vaccine X*, 2019, PMID: 31384749; Ercolano et al, *Nat Comms*, 2021, PMID: 33953160; Hams et al, *J Immunol*, 2013, PMID: 24166975; Miller et al, *JCI Insight*, 2021, PMID: 34699382). Cells isolated from the peritoneum were immunophenotyped and these cells were immunophenotypically ILC2 cells. These data can be found in **Extended Data Figure 1**. Additional information regarding the isolation of ILC2 cells can be found in peer-reviewed publications by Neill DR et al., *Nature*, 2010, Bruce et al, *J Clin Invest*, 2017, and Bruce et al, *Blood Adv*, 2022.

Reviewer 1, Remark 8, Part B. “Figure 1C: how were cells purified? Based on which parameters? Are ILC1 derived from IL-25 stimulated mice bona fide ILC1?”

Again, we apologize for the confusion about the ILC1 cells. We are not generating ILC1 cells directly from IL-25-stimulated mice (these are ILC2 cells). To summarize, ILC2s were elicited via four days of IP administration of recombinant mouse IL-25 as described above. The cells are subsequently purified by negative selection with a MACS column using biotinylated antibodies against a cocktail of lineage antibodies that are specific for T, B and myeloid cells. These cells were then cultured for 6 days in complete media supplemented with IL-7 and IL-33, with the media changed every 2 days. As is mentioned above, this technique has been previously reported to yield highly pure ILC2s in the work of: Neill DR et al, *Nature*, 2010, PMID: 2862165; Bruce et al, *J Clin Invest*, 2017, PMID: 28375154 and Bruce et al, *Blood Adv*, 2022, PMID: 34619767. **Figure 1C** shows the distribution of H3K4me3 signal in ILC2 cells from IL-

25 treated animals at transcriptional start sites of genes differentially marked with H3K4me3 in mouse ILC1 and ILC2 cells. TSSs specifically marked with H3K4me3 in either ILC1 or ILC2 were identified by reanalyzing previously published ILC1 and ILC2 (small intestine) H3K4me3 data (Gury-BenAri, *Cell*, 2016). These ILCs were CD127⁺ and from RORgt-GFP reporter mice, ILC1s were sorted RORgt⁺NKp46⁺, and ILC2s were RORgt⁻KLRG-1⁺. Please see **Extended Data Fig. 1c-e** and our methods section for the description of the differential and clustering analysis used to identify the set of TSSs associated with each cell type.

Reviewer 1, Remark 8, Part C. “Figure 1E: cells gated as lineage neg/ ST2 pos? or show ST2 expression? ST2 is mentioned in legend but not shown in graphs.”

Please refer to **Extended Data Figure 1A**, which provides the gating strategy for the representative flow cytometric histograms shown in **Figure 1E**. This scheme depicts the selection of single cells from the mesenteric lymph nodes and peritoneal lavage that are negative for the live-dead discrimination and that express ST2 but do not stain positive for our Lineage cocktail, which contains antibodies against CD3, B220, CD11b, TER-119, and Gr-1.

Reviewer 1, Remark 8, Part D. “Extended data Figure 1C-E: please clarify R1 and R2.”

We apologize for the lack of clarity in the figure legend for **Figure 1**. We have revised the figure legend for **Extended Data Figure 1C-E** to indicate that R1 and R2 refer to replicate 1 and replicate 2 of the (Gury-BenAri, *Cell*, 2016) H3K4me3 ChIP analyses.

Reviewer 1, Remark 9. “Figure 2C: an alternative explanation to plasticity of ILC2 could be that the 0.3% of ILC in the pre-transplant population have proliferated and seeded the post-transplant population. Assuming exponential growth, only a few rounds of division are needed. How was this excluded?”

We thank the reviewers offering this alternative interpretation of the data. This question led us to perform additional experiments to explore this important point. In our ex vivo expansion experiments, ILC1-like cells represent 0.2-0.5% of the final cell population. Thus, these cells would need to proliferate remarkably to outcompete the overwhelming number of ILC2 cells infused. To directly evaluate this hypothesis, we generated pcILC2 cells, which have an ATAC signature similar to the day 6 ILC1-like cells, marked these cells, and infused them with ILC2 cells, marked in a different manner in a ratio of 1:50 (2% contaminating population). If the contaminating cells were actually leading to the cells characterized by multiome analysis on day 21 post-HSCT, we predicted that the pcILC2 would greatly outcompete the ILC2 cells by three weeks post-transplant. However, our data indicate that this was not the case. We detected pcILC2 cells and ILC2 cells at almost the exact ratio as when they were infused. These data suggest that cytokine treated cells that are skewed to be ILC1-like do not outcompete ILC2 cells in vivo. These data have been added to **Extended Data Figure 4**, and we have inserted new text describing these experimental findings at/on lines **79-97**.

Reviewer 1, Remark 10. *“Figure 3L, M: Survival advantage of mice that received ILC2 not significant in contrast to previous publications (Bruce et al, J Clin Invest 2017, Blood Adv, 2021). Please clarify. In the text (lines 167-168) it is stated that ‘... co-administration of ILC2 significantly diminished clinical GvHD scores and improved survival....’ but this is not supported by the data.”*

We apologize for the misstatement and have removed the reference to statistical significance from the analysis of this experiment. This experiment was not powered to compare the outcome of ILC2 infusion on the development of acute GvHD after allo-HSCT and compare this to the outcome found in our manuscripts published 7 years ago. Further, the experiment was not powered to demonstrate that ILC2 infusion improved the outcome of mice after allo-HSCT compared to recipients receiving T cells without ILC2 cells. Rather, this experiment was performed and powered to ask whether pcILC2 cells are impaired in their ability to prevent GvHD. Indeed, we found that pcILC2 cells unable to mitigate the development of aGvHD. We would also emphasize that there was not a significant difference in the survival of mice receiving ILC2 cells + T cells in this manuscript compared to the results described in our earlier JCI publication despite marked differences in the current and previous work regarding where the mice were procured, where the experiments were performed and the differences in the age of our mouse facility given the differences in the microbiota in our current mice compared to those used more than seven years ago.

Reviewer 1, Remark 11. *“In lines 215-222 it is concluded that NR4A2 and FLII are novel regulators of ILC2 plasticity. However, functional experiments e.g. knockout experiments to demonstrate a functional relation to back this statement are lacking. Also, it is unclear whether the authors refer to mouse or human ILC here.”*

We agree with the reviewers that the prior language regarding the function of these candidate regulators implied results of experiments that we have yet to perform. We now indicate that the increase in the abundance of the RNA for these genes as well as increased chromatin accessibility as sites that contain their DNA binding motifs merely suggests these transcriptional regulators could potentially play a role in the plasticity of ILC2 cells in the current manuscript.

Laurie et al responses to Reviewer #2

Reviewer 2, Remark 1. *“That ILC2 fail to reconstitute in allo-HSCT in humans is, however, incorrect. Ref 6 describe that ILC2 do reconstitute but at a slower pace than the other ILC subsets. There is no evidence in humans that ILC2 protect against acute GVHD. Ref 6 seems to indicate that ILC3 are mediating this effect.”*

Our group and others have found that ILC2 cells remain substantially decreased for more than one year after allo-HSCT demonstrating an impaired ability to reconstitute to pre-transplant levels after this procedure (Munneke et al, *Blood*, 2014, PMID: 24855210; Vely et al, *Nat*

Immunol, 2016, PMID: 27618553; Piperoglou et al, *J Leuk Biol*, 2022, PMID: 33847423; Jarosch et al, *Cell Rep Med*, 2023). We have altered the manuscript to indicate this. We are unaware of data that indicates that the infusion of ILC3 cells can mitigate the incidence or severity of acute GvHD. Reference 6 is a publication from our collaborator Alan Hanash from the time when he was in the van den Brink laboratory. Dr. Hanash found that IL-22 protected against the loss of intestinal stem cells that occurs during GvHD. The ability of IL-22-Fc to protect from aGvHD was recently evaluated in humans (Ponce et al, *Blood*, 2023). In this study, patients with acute GvHD of the lower GI tract were treated with corticosteroids administered together with F-652, a recombinant IL-22 dimer. In that study, the CR rate for treatment was 48%, which is similar to the published data when patients are treated with steroids alone. F-652 was much more effective in combination with corticosteroids in the treatment of patients with lower stage acute GvHD. Thus, (1) it is currently not clear if IL-22 is an effective treatment for acute GvHD of the lower GI tract and (2) there are no data to indicate impaired reconstitution of ILC3 cells after allo-HSCT.

Reviewer 2, Remark 2. *“Whereas the data in the mouse model are generally supporting the interpretation of the authors. However, it is difficult to judge whether the pcILC2 are really derived from the ILC2 and not from ILC1s contaminating the ILC2 population and expanded in the ILC1 permissive condition. The authors need to provide information of the purity of the starting cell population and the expanded cells.”*

Please see Response #9 to Reviewer One’s similar comment above.

Reviewer 2, Remark 3. *“It is not possible to judge the human data. No information is given of how the human ILC2s are isolated and what the phenotype is of these cells before and after culture in ILC2 or ILC1 permissive conditions.”*

We apologize for providing an incomplete description of the isolation and phenotype of our human ILC2 cells and are grateful for the opportunity to include these data in our revised manuscript. Additional details of isolation of ILC2s from healthy human donor peripheral blood lymphocytes can now be found on page 20, with a description of the expansion of these cells on page 21. We have added more information regarding the clinical characteristics of our HSCT patients on pages 22 and 23 and have provided new text describing the phenotyping of these cells has been inserted and highlighted on lines **148-165** and the corresponding data have been inserted as new **Extended Data Figure 5a-b**. To be clear, given the paucity of cells in the peripheral blood, we did not isolate ILC2 cells for this analysis. As described in response to Reviewer 1, question 5, we identified a set of genomic regions with increased ATAC signal specific to human ILC2 and pcILC2 cells. We quantified ATAC signal at these regions from the circulating mononuclear cells in the bloodstream in healthy controls and patients post allo-HSCT with and without acute GvHD. In an effort to focus our evaluation on innate lymphoid cells (which are solely ILC2 cells in humans), we filtered out cells that expressed MPO, CD19, CD20,

CD22 and CD3 RNA. Despite this approach to exclude myeloid, B and T cells, the overall results were unchanged.

Laurie et al responses to Reviewer #3

Reviewer 3, Remark 1. *"As a major recommendation, I suggest including a section in the manuscript dedicated to data availability. This section should specify the GEO repositories, where the generated data can be accessed."*

We thank Reviewer 3 for highlighting the missing data availability section and have inserted and highlighted the source of the data on lines **754-759** of the revised manuscript presented here.

Reviewer 3, Remark 2. *"Additionally, as a minor suggestion, I recommend expanding the GitHub repository's readme file. Specifically, consider adding an introductory chapter that succinctly summarizes the purpose of the GitHub repository and its connection to the associated research paper."*

We have added an about statement to the top of the GitHub (<https://github.com/j-foster2/GvHD>) that describes the purpose of the repository and links it to the manuscript.

Again, we thank the reviewers for their time and insightful comments. We strongly believe that the current version of the manuscript has been substantially enhanced because of the robust and inciteful previous review. We hope the revised version of this manuscript is suitable for publication in Nature Communications.

Sincerely yours,

Jonathan S. Serody, MD

Elizabeth Thomas Professor of Medicine, Microbiology and Immunology
Associate Director for Translational Science, Lineberger Cancer Center
Chief Division of Hematology in the Department of Medicine
Director, Cellular Therapy Program, UNC Health Care
University of North Carolina at Chapel Hill
Chapel Hill, NC, USA

REVIEWERS' COMMENTS

Reviewer #1 (Remarks to the Author):

Thank you for your extensive revisions, most of the points that I raised have been answered to and I agree with the authors that the manuscript has improved significantly. However, 2 important points remain (that can be resolved by rewording):

1. The authors still state that poor ILC2 recovery after allogeneic HCT is due to conversion to ILC1-like cells (page 1 lines 35-37: 'These data demonstrate [...] that ILC2s poorly repopulate the lower GI tract after allo-HSCT in part due to conversion to [...] ILC1-like [cells], providing novel insights into the contribution of ILC plasticity to the impaired reconstitution of ILC2 after allo-HSCT.' and page 9 lines 267-269: 'In summary, we demonstrate that loss of ILC2 after transplantation is associated with conversion of those cells to an ILC1-like phenotype').

If it would be only a matter of plasticity then one would expect an expansion of ILC1. However, in their rebuttal the authors state that that is not the case (see Reviewer 1 Remark 1: 'ILC1 were reduced, suggesting that ILC generation may be globally affected post allo-HSCT'). The data convincingly show that the large majority of transfused, GFP+ ILC2 indeed convert to ILC1-like cells. However, this is the majority of probably what is still a small population. In other words: the data show proportions only and there is no data showing an increase in absolute numbers of ILC1 after transplantation. I agree with the authors that it is very likely that 'ILC generation is globally affected'. That would also fit with the observation in human allogeneic HSCT recipients of delayed recovery of the total ILC pool and of ILC1 and ILC2 (Munneke et al., Blood 2014). While it is clear that transferred ILC2 are plastic in this mouse model and it is very possible that plasticity occurs also in humans, it is important to realize that if it would all be plasticity, then one would expect normalization of circulating numbers of ILC1 in the months after transplantation and that is not the case. It would strongly recommend to change '..in part due to conversion..' (page 1 lines 36-37) into '..and convert to..'.

In fact, the new observation that ILC2 recovery depends on the presence of donor T cells in the syngeneic setting and the absence of a pro-inflammatory environment is very interesting and merits a more prominent place in the abstract.

2. My second concern relates to the statement that '.. the co-administration of ILC2s diminished clinical GvHD scores and improved survival..(Fig 3m, 3n)' (page 7 lines 199-201). When looking at figures 3m and 3n one can only conclude that clinical GvHD score lines are fully overlapping (fig 3m) and that adding ILC2 to BM+T cells did not significantly improve survival compared to mice who received BM+T cells without ILC2 (fig 3n). I understand from the rebuttal that in the current study ILC2 were not titrated to concentrations sufficient to prevent or mitigate GvHD because that was not the scope of the study. That is plausible and totally reasonable, but the text should be rephrased.

Reviewer #2 (Remarks to the Author):

The revised paper of Serody et al has been improved but the human part remains problematic. They now have changed the claim that there are no ILC2 after HSCT to that these cells show a delayed recovery after HSCT. This may be true if we look only at PBMC but as long as the recovery in tissues is unknown one should be careful with this claim.

To my surprise the authors responded to my remarks on reference 6 with text referring to another paper. Reference 6 is a publication by Munneke et al. and not by Hanash et al. Munneke has shown that the presence of host-derived NCR1+ILC3 in PBMC after myoablative conditioning and of donor-derived NCR1+ILC3 after HSCT was associated with a reduced susceptibility for GvHD. This suggests although not prove that NCR1+ILC3 are implicated in protection against damage caused by the conditioning and the transplantation. The authors have to include a correct discussion of

this paper pointing to the possibility that in humans ILC3s and not ILC2s are the cells protecting against GvHD.

As referee 1 pointed out the level of ILC2s in the adult human intestine is either absent or very small and this referee questioned the relevance of the data obtained with PBMC. The authors cite several papers that would show that ILC2s are present in the human intestine. These papers confirm, however, that ILC2s are hard to detect in human intestine. In Groc et al. ILC2 were not found (Fig 1A of this paper) In Krämer et al ILC2 were found to be a very minor population in all parts of the alimentary tract and in Moller et al ILC2s were found in the intestine of children but not or only sporadically in intestine of adults. The authors therefore cannot exclude that their data in PBMC have nothing to do with what happens in the intestine. The paper of Jarosch et al which the authors cite as showing that ILC2s are present in healthy intestine but absent in patients with acute GvHD the is not very informative because of the absence of a detailed description of how these cells were characterized and, more importantly, the absence of statistical data. This is important as ILC2s are very hard to detect in human intestines.

The authors show that pILC2 that expand in mice are not the result of prior contamination with ILC1-like cells. However, this was not excluded in the experiment depicted in Fig. 4. The ILC2s were not purified with rigorous sorting using a flow cytometer but with a RosetteSep separation. The possible contamination with ILC1s and ILC3s is unknown and the possibility that their samples cultured with the proinflammatory cytokines (which may also revert contaminating ILC3s) contain expanded ILC1s cannot be excluded diminishing the validity of the data.

Minor remarks:

The claim that their paper is the first to evaluate ILC2 signatures in human PBMC at the chromatin level is incorrect. Such signatures albeit in a different context were published by Stadhouders and collaborators (PMID: 29486229, PMID: 33514640)

The patient selection and information was described in lines 538 – 548 and not in 572-582 as stated in the rebuttal. The conditioning regimen was not mentioned. This is important because Vely et al has shown reconstitution of ILCs after transplantation only in myeloablative conditioned patients.

The Authors state “We have added more information regarding the clinical characteristics of our HSCT patients on pages 22 and 23 and have provided new text describing the phenotyping of these cells has been inserted and highlighted on lines 148-165 and the corresponding data have been inserted as new Extended Data Figure 5a-b” I couldn’t find this information in the indicated lines.

Reviewer #3 (Remarks to the Author):

I am completely satisfied with the responses provided by the authors and the modifications made to both the manuscript and the GitHub repository.

29 May 2024

Nature Communications
London, UK

We thank the Reviewers for their time and thoughtfulness in providing additional remarks and insights that will allow clarification of key points in our manuscript. The following are a series of point-by-point responses to address the remaining referee queries that accompany new changes made to our manuscript which have been marked in track changes and line/page numbers have been indicated below. Our responses to the most recent queries by the reviewers are in blue font.

REVIEWERS' COMMENTS

Reviewer #1 (Remarks to the Author):

Thank you for your extensive revisions, most of the points that I raised have been answered to and I agree with the authors that the manuscript has improved significantly. However, 2 important points remain (that can be resolved by rewording):

1. “The authors still state that poor ILC2 recovery after allogeneic HCT is due to conversion to ILC1-like cells (page 1 lines 35-37: ‘These data demonstrate [...] that ILC2s poorly repopulate the lower GI tract after allo-HSCT in part due to conversion to [...] ILC1-like [cells], providing novel insights into the contribution of ILC plasticity to the impaired reconstitution of ILC2 after allo-HSCT.’ and page 9 lines 267-269: ‘In summary, we demonstrate that loss of ILC2 after transplantation is associated with conversion of those cells to an ILC1-like phenotype’).

If it would be only a matter of plasticity then one would expect an expansion of ILC1.”

We agree with Reviewer #1 that there may be more than one mechanism for the poor reconstitution of ILC2 cells after HSCT. Please see lines 39-42 on page 1 (bottom of summary paragraph/abstract) as well as page 9, lines 270-271, for changes to the discussion that clarify our conclusion that the plasticity of ILC2s is likely only one of several potential mechanisms for the poor reconstitution of these important cells after allo-HSCT. Finally, while we agree with this reviewer’s comments regarding other mechanisms that may mediate the markedly reduced numbers of ILC2 cells post allo-HSCT, the detection of increased numbers of ILC1 cells as a marker of-plasticity is challenging to interpret as the ability to identify plastic ILC2 cells is based on their detection as a reasonable fraction of the total number of ILC1 cells. Although clearly not conclusive, it is interesting that there appears to be a relationship between decreased ILC2 cells (%) and increased ILC1 cells in grades 2-4 aGvHD in the data from (Fig S2) Jarosch et al.

“However, in their rebuttal the authors state that that is not the case (see Reviewer 1 Remark 1: ‘ILC1 were reduced, suggesting that ILC generation may be globally affected post allo-HSCT’). The data

convincingly show that the large majority of transfused, GFP⁺ ILC2 indeed convert to ILC1-like cells. However, this is the majority of probably what is still a small population. In other words: the data show proportions only and there is no data showing an increase in absolute numbers of ILC1 after transplantation. I agree with the authors that it is very likely that ‘ILC generation is globally affected’. That would also fit with the observation in human allogeneic HSCT recipients of delayed recovery of the total ILC pool and of ILC1 and ILC2 (Munneke et al., Blood 2014). While it is clear that transferred ILC2 are plastic in this mouse model and it is very possible that plasticity occurs also in humans, it is important to realize that if it would all be plasticity, then one would expect normalization of circulating numbers of ILC1 in the months after transplantation and that is not the case. It would strongly recommend to change ‘..in part due to conversion..’ (page 1 lines 36-37) into ‘..and convert to’....

We agree with the reviewer and have made this change in the manuscript text on page one at the bottom of our opening summary paragraph (abstract).

“In fact, the new observation that ILC2 recovery depends on the presence of donor T cells in the syngeneic setting and the absence of a pro-inflammatory environment is very interesting and merits a more prominent place in the abstract.”

We appreciate this point and have added the following additional sentence into the updated abstract (including a few other small changes proposed by the editorial team) on page 1, lines 34-35, that makes this observation more clear such that it now reads: ‘... Whereas ILC2s protect against acute graft-versus-host disease- (aGVHD) mediated morbidity and mortality, infusion of ILC2s exposed to proinflammatory cytokines accelerates aGvHD. *Interestingly, murine ILC2 reconstitution post-HSCT is decreased in the presence of alloreactive T cells.* Finally, circulating peripheral blood mononuclear cells from human HSCT patients with aGvHD have an altered ILC2-associated chromatin landscape compared to transplanted controls...’

2. “My second concern relates to the statement that ‘.. the co-administration of ILC2s diminished clinical GvHD scores and improved survival..(Fig 3m, 3n)’ (page 7 lines 199-201). When looking at figures 3m and 3n one can only conclude that clinical GvHD score lines are fully overlapping (fig 3m) and that adding ILC2 to BM+T cells did not significantly improve survival compared to mice who received BM+T cells without ILC2 (fig 3n). I understand from the rebuttal that in the current study ILC2 were not titrated to concentrations sufficient to prevent or mitigate GvHD because that was not the scope of the study. That is plausible and totally reasonable, but the text should be rephrased.”

We appreciate this comment and have rephrased the text on page 7 to include the statement that our differences are not significant. The original language used in the previous draft of the manuscript read on page 7, lines 202-203: “As we have previously published, the co-administration of ILC2s diminished clinical GvHD scores and improved survival; strikingly, the pILC2s failed to mitigate aGVHD and instead led to an increase in disease mortality (Fig. 3 m, n)” We have now altered this text to state: “Strikingly, pILC2s failed to mitigate aGvHD and instead led to an increase in disease mortality compared to mice receiving T cells + ILC2s.”

Reviewer #2 (Remarks to the Author):

“The revised paper of Serody et al has been improved but the human part remains problematic. They

now have changed the claim that there are no ILC2 after HSCT to that these cells show a delayed recovery after HSCT. This may be true if we look only at PBMC but as long as the recovery in tissues is unknown one should be careful with this claim.”

We appreciate this point and have modified the text in our discussion on lines 246-247 on pg. 8 to reflect that the ability to quantify ILC2 reconstitution after allo-HSCT is predominantly limited to circulating cells.

“To my surprise the authors responded to my remarks on reference 6 with text referring to another paper. Reference 6 is a publication by Munneke et al. and not by Hanash et al. Munneke has shown that the presence of host-derived NCR1+ILC3 in PBMC after myoablative conditioning and of donor-derived NCR1+ILC3 after HSCT was associated with a reduced susceptibility for GvHD. This suggests although not prove that NCR1+ILC3 are implicated in protection against damage caused by the conditioning and the transplantation. The authors have to include a correct discussion of this paper pointing to the possibility that in humans ILC3s and not ILC2s are the cells protecting against GvHD.”

We have included new language in our discussion on page 9, lines 270-274, to address the work from the van der Brink laboratory and most recently the Hanash laboratory, stating that our findings complement previous work indicating impaired reconstitution of NCR+ ILC3 cells, which have also been shown to mitigate acute GvHD of the GI tract via the expression of IL-22.

“As referee 1 pointed out the level of ILC2s in the adult human intestine is either absent or very small and this referee questioned the relevance of the data obtained with PBMC. The authors cite several papers that would show that ILC2s are present in the human intestine. These papers confirm, however, that ILC2s are hard to detect in human intestine. In Groc et al. ILC2 were not found (Fig 1A of this paper). In Krämer et al ILC2 were found to be a very minor population in all parts of the alimentary tract and in Moller et al ILC2s were found in the intestine of children but not or only sporadically in intestine of adults. The authors therefore cannot exclude that their data in PBMC have nothing to do with what happens in the intestine. The paper of Jarosch et al which the authors cite as showing that ILC2s are present in healthy intestine but absent in patients with acute GvHD the is not very informative because of the absence of a detailed description of how these cells were characterized and, more importantly, the absence of statistical data. This is important as ILC2s are very hard to detect in human intestines.”

We agree with this point from Reviewer #2 and for the reasons stipulated have not attempted to comment on how our data reflects on the presence/absence of ILC2 cells in the human GI tract. To address previous reviewer comments, we analyzed the data from Jarosch et al. as these constituted the only data set available that enabled evaluation of ILCs in the GI tract in patients with GvHD. As commented by the reviewer, we were not able to draw firm conclusions from these data given the paucity of samples described in that manuscript. We also believe that the presence of ILC2s in the GI tract is likely disease-associated, as this compartment seems to be more robust in inflammatory settings (e.g., IBD) vs. others (steady state GI). However, conclusive demonstration of these cells in the GI tract, necessitates a clinical trial in which donor ILC2 cells are infused into patients with treatment-resistant acute GvHD of the lower GI tract, and the infused cells are identified using spatial/single cell transcriptomics. We are in the process of generating an IND for the use of donor ILC2 cells for this indication, and-biological correlatives from that trial should address this concern.

“The authors show that pILC2 that expand in mice are not the result of prior contamination with ILC1-like cells. However, this was not excluded in the experiment depicted in Fig. 4. The ILC2s were not purified with rigorous sorting using a flow cytometer but with a RosetteSep separation. The possible contamination with ILC1s and ILC3s is unknown and the possibility that their samples cultured with the proinflammatory cytokines (which may also revert contaminating ILC3s) contain expanded ILC1s cannot be excluded diminishing the validity of the data.”

Although we did not include the data in our original submission, we have flow cytometry data that support the specificity of the ILC2 chromatin accessibility pattern detected at the chromatin level by ATAC sequencing in Figure 4b. Below, we demonstrate that at the protein level human ILC2 cells isolated by RosetteSep from healthy human donors are devoid of an ILC1-like signature unless they are exposed to the proinflammatory cytokines IL-18, IL-1 β , and IL-12.

Gated on live, lineage negative singlets
Lineage = CD2, CD3, CD14, CD16, CD19, CD56, CD235a

On the transcriptomic level, we profiled human ILC2s expanded from healthy donors. These data are described in a separate manuscript that is currently under review at *Blood Advances* (submitted May 2024, manuscript number ADV-2024-013609). These data indicate that the lineage signature score (generated from two other published datasets describing ILC1/2/3s) of our ex vivo expanded human

ILC2s are transcriptomic similar to published ILC2 transcriptional signatures but not with ILC1 or ILC3/ILCp, with the latter the least similar to our ILC2 cells by this metric.

UNC – University of North Carolina at Chapel Hill (Serody Lab)
 UMN – University of Minnesota (Blazar Lab)
 Ercolano et al, J Leuk Biol, 2020

Wang et al, Biorxiv, 2021, doi.org/10.1101/2021.04.20.440368
 Mazurana et al, Cell Res, 2021

FROM BLOOD ADV-2024-013609

The protein level data from the Blood Advances manuscript show that although hILC2 cells make up the majority of the cells in the final ex vivo expanded product described in this manuscript, we detected a very small contaminating population of ILC1s but not ILC3s. The contaminating population was significantly decreased by culturing with IL-4 or anti-IL-12 mAb consistent with these cells being ILC1-like and not ILC3-like cells. Given the significant increase in ILC1-like cells in response to proinflammatory cytokine exposure as described herein in Figure 4 and the flow figure provided with this rebuttal, we believe the ATAC-seq data presented in this manuscript overwhelmingly derive from ILC2 vs pcILC2 cells and not ILC1 or ILC3 cells.

Reviewer #2 (Minor remarks):

“The claim that their paper is the first to evaluate ILC2 signatures in human PBMC at the chromatin level is incorrect. Such signatures albeit in a different context were published by Stadhouers and collaborators (PMID: 29486229, PMID: 33514640).”

We apologize for the confusion; our original statement was meant to indicate that our study is the first published evaluation of a human ILC2 signature in the peripheral blood using single-cell ATAC data. Nonetheless, we appreciate the studies by Stadhouers et al (*J Allerg Clin Immunol*, 2018, PMID 29486229) and van der Ploeg et al (*Sci Immunol*, 2021, PMID 33514640) and have removed the “first to evaluate” claim from the finalized draft of our manuscript.

“The patient selection and information was described in lines 538 – 548 and not in 572-582 as stated in the rebuttal. The conditioning regimen was not mentioned. This is important because Vely et al has shown reconstitution of ILCs after transplantation only in myeloablative conditioned patients.”

We appreciate Reviewer #2’s detailed attention to our manuscript and confirm that the additional information describing the selection and characteristics of the human patients was, in fact, on lines 538-548; we apologize for this error. Additionally, we recognize the importance of including detailed information regarding the conditioning regimens used in preparing these individuals for transplant given the work of Vely et al. highlights the impact of myelo- vs. non-myeloablative conditioning on the reconstitution of innate lymphoid cells after HSCT. We have now included these additional details on lines 550-554, in which we describe that, of the patients who remained stable after allo-HSCT, two out of three received myeloablative conditioning regimens while one underwent non-myeloablative conditioning. Similarly, of the patients who developed acute GVHD in the first four months following their allo-HSCT, two out of three received myeloablative conditioning regimens while one underwent non-myeloablative conditioning.

“The Authors state “We have added more information regarding the clinical characteristics of our HSCT patients on pages 22 and 23 and have provided new text describing the phenotyping of these cells has been inserted and highlighted on lines 148-165 and the corresponding data have been inserted as new Extended Data Figure 5a-b” I couldn’t find this information in the indicated lines.”

The reviewer is correct in pointing out that we had not inserted highlighted text at that site, and we apologize for the lack of clarity regarding the inclusion of addition of the human data. Please note that the updated patient characteristics can now be found on page 16, lines 547-554 (what we had added in the ‘original’ rebuttal is on what is now pp. 17-18 regarding the text in methods on patient selection, IRB approval, and clinical characteristics). Additionally, we have added text on page 7, lines 212-214 that describes the gating and identification of peripheral blood ILC2 cells from healthy donors after isolation from TRIMA cone by Rosette Separation in Extended Data 5a, Extended Data 5b shows representative flow data showing our strategy for the identification of ex vivo expanded ILC2s. Note that we have also provided additional protein level data characterizing human ILC2s and pcILC2s in response to comment 3 above that provides complementary phenotyping data.

Reviewer #3 (Remarks to the Author):

“I am completely satisfied with the responses provided by the authors and the modifications made to both the manuscript and the GitHub repository.”

We are grateful for the opportunity to respond to Reviewer #3’s previous comments and are glad we were able to address them completely.

Again, we thank the reviewers for their time commitment and thoughtful and insightful comments. We strongly believe that the current version of our manuscript having addressed these concerns is suitable for publication in *Nature Communications*.

Sincerely yours,

Jonathan S. Serody, MD

Elizabeth Thomas Professor of Medicine, Microbiology and Immunology
Associate Director for Translational Science, Lineberger Cancer Center
Chief Division of Hematology in the Department of Medicine
Director, Cellular Therapy Program, UNC Health Care
University of North Carolina at Chapel Hill
Chapel Hill, NC, USA